# grandR: a comprehensive package for nucleotide conversion RNA-seq data analysis

Teresa Rummel [1,3], Lygeri Sakellaridi [1,3] & Florian Erhard [1,2] ✉

Metabolic labeling of RNA is a powerful technique for studying the temporal dynamics of gene expression. Nucleotide conversion approaches greatly facilitate the generation of data but introduce challenges for their analysis. Here we present grandR, a comprehensive package for quality control, differential gene expression analysis, kinetic modeling, and visualization of such data. We compare several existing methods for inference of RNA synthesis rates and half-lives using progressive labeling time courses. We demonstrate the need for recalibration of effective labeling times and introduce a Bayesian approach to study the temporal dynamics of RNA using snapshot experiments.

The RNA expression level of a gene is governed by the interplay of RNA synthesis and degradation. While RNA-seq can easily obtain transcriptome-wide snapshots of gene expression profiles in a single experiment, it remained difficult to directly measure the temporal dynamics of gene regulation as consequences of changes in the rates of RNA synthesis and degradation, e.g. due to external stimuli.

To overcome this limitation, techniques involving metabolic labeling of RNA have been developed. Metabolic RNA labeling uses 4-thiouridine (4sU) or other nucleoside analogs, which are introduced into living cells and incorporated into nascent RNA. First approaches physically separated labeled and unlabeled RNA by thiol-specific biotinylation and affinity purification and sequenced separate libraries of these fractions. This approach has been used to study RNA processing[1], transient RNA expression[2], kinetics of RNA polymerases[3] or the dynamics of RNA expression[4]. Existing protocols for purification of labeled RNA are highly laborious and require substantial amounts of RNA[5]. In addition, contamination with background total RNA in the labeled RNA fraction must be controlled for[4], and normalization is challenging[6].

Recently, several approaches have been proposed that circumvent the purification[7–9]: Before sequencing, RNA is treated with chemical agents to specifically convert 4sU into cytosines or cytosine analogs. Thus, labeled and unlabeled RNAs can in principle be differentiated based on T-to-C mismatches in sequencing reads without the need to physically purify labeled RNA. A major advantage of the nucleotide conversion approach, aside from lower requirements of starting material and a simplified experimental workflow, is that it can

be combined with more specialized protocols, e.g. to profile transcription start sites[10] or ribosome occupancy[11]. Furthermore, we and others have combined 4sU conversion with single cell RNA-seq to study the heterogeneity of gene regulation[5,12–14].

A limitation of 4sU conversion approaches is that concentrations of 4sU that are tolerated by cells commonly only replace 1 in 40 uridines by 4sU[7]. Thus, a considerable number of reads originating from labeled RNA does not cover any 4sU incorporation site. The percentage of such reads is in the order of 20-80% and depends on the ratio of 4sU and normal uridine available for incorporation into nascent RNA, the read length, and the uridine content of RNA. The pioneering studies employing 4sU conversion exclusively focused on T-to-C reads. Despite underestimating labeled RNA, T-to-C reads alone can be used to estimate unbiased RNA half-lives in pulse-chase experiments[7] and to detect rapid changes of transcription upon drug treatment or acute depletion of transcription factors[3,15,16].

We previously proposed a statistical solution to quantify labeled and unlabeled RNA without bias due to limited 4sU incorporation: By using a binomial mixture model, our GRAND-SLAM approach provides unbiased estimates of the percentage of labeled RNA per gene and its posterior distribution[17]. The posterior represents uncertainties in the quantification, mainly due to the scarcity of 4sU incorporation events, and has so far been used to filter out genes with inaccurate quantification[12]. Importantly, however, our Bayesian framework in principle allows to take these uncertainties further to downstream analyses such as estimation of RNA kinetics or gene expression changes.

[1]Institute for Virology and Immunobiology, University of Würzburg, Versbacher Str. 7, 97078 Würzburg, Germany. [2]Faculty for Informatics and Data Science, University of Regensburg, Bajuwarenstr. 4, 93053 Regensburg, Germany. [3]These authors contributed equally: Teresa Rummel, Lygeri Sakellaridi. ✉e-mail: Florian.Erhard@informatik.uni-regensburg.de

GRAND-SLAM is designed for the primary processing of nucleotide conversion RNA-seq data, i.e. to estimate the percentage of labeled RNA among the total RNA for each gene. Estimating RNA half-lives[3,18] or uncovering short-term gene regulation[12,15] require additional analysis steps. For experiments involving affinity purification of labeled RNA, several software packages including pulseR[19], INSPEcT[20] and DRiLL[4] have been developed for such downstream analyses. However, all these packages are designed to work with separate sequencing libraries corresponding to the labeled and unlabeled fraction of a sample, and the models implemented in these packages are blind towards the challenges that come along with nucleotide conversion approaches. As a notable exception, pulseR has later been adapted for analyzing nucleotide conversion data by adding an additional nuisance parameter for the limited labeling efficiency[21]. However, none of these methods take advantage of the output of the mixture modeling approach as implemented in GRAND-SLAM or the fact that uncertainties of the percentage of labeled RNA can be estimated for nucleotide conversion approaches.

To close this gap, we here present grandR, an R package to facilitate analyses of nucleotide conversion RNA-seq experiments and taking full advantage of the uncertainty estimates. It includes new methods for quality control and recalibrating labeling times. grandR implements several methods to estimate RNA synthesis and degradation rates from progressive labeling experiments that have been applied previously by us and others[3,17–19,22]. Here, we compare these methods and show that the most accurate results are obtained by directly utilizing the posteriors from GRAND-SLAM to estimate the kinetic model. Furthermore, we propose a Bayesian hierarchical model to dissect the mode of gene regulation from snapshot experiments. To facilitate collaborative work and exploratory data analysis, grandR provides a comprehensive web-based data visualization and exploration tool.

## Results

### grandR overview

grandR is designed as a comprehensive and easy-to-use toolkit for all types of nucleotide conversion RNA-seq data such as SLAM-seq[7], Timelapse-seq[8] or TUC-seq[9]. The inputs for grandR are in principle matrices (genes × samples) of (i) total read counts and (ii) information derived from metabolic RNA labeling (e.g., read counts of labeled RNA

or the percentage of labeled RNA). grandR includes several functions to read data into a common internal data structure, and most of the implemented methods are therefore agnostic how raw data has been preprocessed. The only exceptions are the Bayesian methods described below, as they require the posterior distributions of the percentage of labeled RNA. We thus recommend using grandR after primary processing by our GRAND-SLAM tool[17], which provides these posterior distributions. We put special emphasis on designing our package to be usable by non-experts in R programming: All data and metadata belonging to a project are stored in a single variable, and there are clearly defined, and expressively named, high-level functions to proceed with analysis pipelines (Fig. 1a). Our workflow advocates (but does not enforce) using systematic sample names to encode metadata (Fig. 1b). After reading data either from the output of GRAND-SLAM, or from count matrices of total and labeled RNA, a typical workflow will (i) filter genes according to user defined criteria, (ii) perform quality control (PCA, toxicity analysis), (iii) normalize data by one of the implemented approaches (size factors from all genes or spike-ins, TPM, FPKM), (iv) perform differential gene expression analysis or kinetic modeling and (v) visualize results. Each of these five steps is accomplished by calling grandR functions (Fig. 1a).

Quality control is important to exclude effects of metabolic RNA labeling on the biology of the cells. Below, we describe the methods we developed for this and that are all implemented in grandR. For differential gene expression, grandR interfaces with DESeq2[23] and the lfc package[24]. P-values and log fold changes can be estimated in a straightforward manner for a single pairwise comparison (e.g., treatment vs. control), or for multiple pairwise comparisons (e.g., treatment vs. control over several time points), either for total, labeled or unlabeled RNA. Testing for more complex designs is also possible based on the likelihood ratio test. grandR implements all kinetic modeling approaches discussed below (pulseR, linear regression after logarithmizing, the non-linear least squares and the Bayesian approaches), it supports both pulse-chase and progressive labeling designs and can also fit non-steady state data (Fig. 1c). The Bayesian hierarchical model for snapshot designs and the temporal recalibrations methods proposed below are also implemented.

For nucleotide conversion RNA-seq data with single cell resolution[12,14], grandR interfaces with the Seurat package[25] and can integrate the metabolic labeling information by different means as

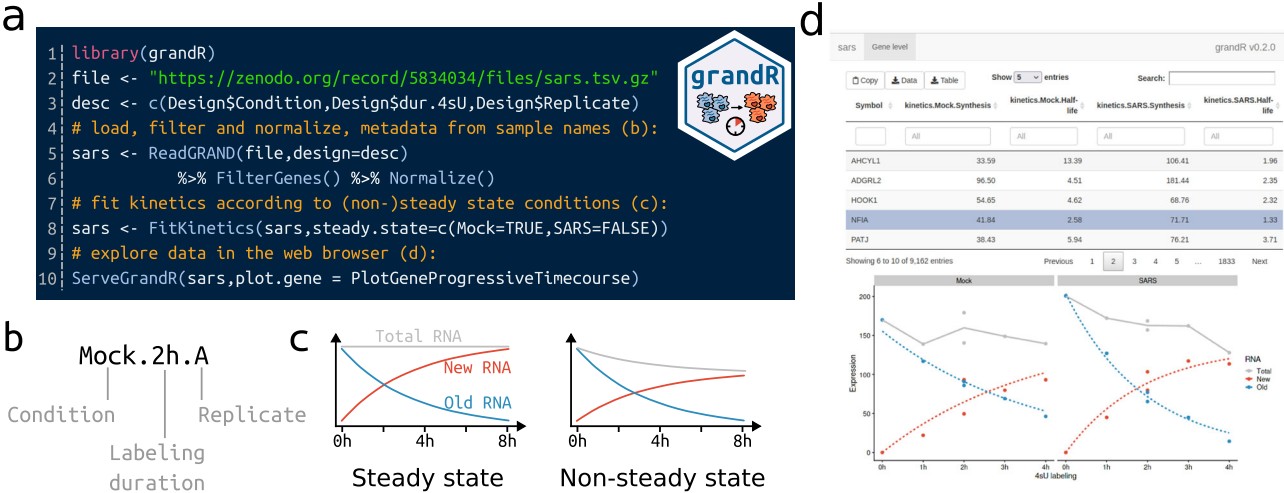

**Fig. 1 | grandR overview. a** Coding example of a grandR project. Self-explanatory high-level commands (blue) load and preprocess data (lines 3-6), and then fit a kinetic model for each gene (line 8). Finally, the interactive web-based tool is started (line 10). Code comments (orange) refer to the other panels. **b** Systematic sample names. When sample names systematically encode metadata in separate fields as shown, grandR can extract these automatically by defining the semantics as shown in lines 3 and 5 in **a**. **c** grandR can fit kinetic models of RNA with or without assuming steady state expression. **d** Web-based data visualization. This interactive graphical user-interface presents a table of analysis results that can be filtered, searched and exported, and experiment-specific visualizations are displayed for individual genes.

described previously[5]. grandR provides tools for visualizations of individual genes and summaries of a data set, which can be used programmatically or via a shiny-based web interface (Fig. 1d). All figures here were generated using grandR, and R notebooks to reproduce all analyses are provided as Supplementary Software 1. Our package comes with extensive documentation, several vignettes describing common workflows and frequently used tasks, and a cheat sheet summarizing the most important functions.

## Quality control reveals impact of long-term labeling on transcription

For nucleotide conversion approaches, sufficient 4sU incorporation into newly synthesized RNA must be achieved to enable accurate quantification. However, labeling with high 4sU concentrations or labeling over extended periods of time affects cell viability[7] or RNA metabolism[26] in a cell type specific manner. Thus, as opposed to standard RNA-seq, nucleotide conversion RNA-seq requires additional quality control steps in the analysis workflow.

In grandR, testing for effects of 4sU on gene expression estimates can be performed by comparing 4sU treated samples against equivalent 4sU naïve control samples. To estimate RNA half-lives, Herzog et al.[7] pre-treated mouse embryonic stem cells (mESCs) with low concentrations of 4sU (100 μM) for a 24 h pulse phase, followed by washing out 4sU and sequencing at several time points during this chase phase. Notably, cell viability was assessed to be ~80% after 24 h. Quality control using grandR revealed that samples treated with 4sU for 24 h and untreated controls segregate in a principal component analysis (PCA, Fig. 2a) and that the p53 pathway was significantly up-regulated, whereas several stress-related pathways were significantly down-regulated (FDR < 0.05, gene set enrichment analysis, Supplementary Fig. 1a). Interestingly, only 447 out of 7215 genes (6.2%) where

significantly (FDR < 5%, DESeq2 Wald test[23]) and strongly (absolute log$_2$ fold change >1[24]) regulated (Supplementary Fig. 1b). Heatmap analysis of these 447 genes also involving the chase timepoints revealed that at least a subset of those genes reverted to their expression level in control cells during the 24 h of 4sU washout (Fig. 2b). Of note, also among the remaining 6,768 genes there were genes with weaker, but consistent up- or downregulation and reversal to the original state in the full time course (Supplementary Fig. 1c, Supplementary Data 1). This reversal was also observed in a PCA where the chase time points gradually moved towards the unlabeled controls (Supplementary Fig. 1d). Heatmap analysis of mESCs with <24 h labeling demonstrated that 4sU induced regulation is virtually absent with 3 h labeling, but already detectable with 12 h labeling (Supplementary Fig. 1e). Taken together these results suggest that a subset of all genes was regulated due to long-term 4sU treatment.

The observed changes in gene expression are due to a change in RNA synthesis or stability. If they are only a consequence of a change in RNA synthesis, monitoring the drop of labeled RNA in the chase phase provides unbiased half-lives. If the observed up- or downregulation of a gene is due to an increase or decrease in RNA stability, the estimated RNA half-life corresponds to the stability under 4sU treatment. If the expression reverts to the state in 4sU naïve cells during the chase phase, the estimated RNA half-life is in between the half-lives in 4sU treated and 4sU naïve cells (see Supplementary Note 1). Thus, if RNA stability plays a role in the up- and down-regulation of genes in 4sU labeled cells, the pulse-chase experiment would over- or underestimate the RNA half-life in 4sU naïve cells, respectively. However, a comparison of half-lives estimated from the 4sU-pulse-chase experiment to half-lives estimated from Actinomycin-D treated mESCs did not show a systematic over- or underestimation of the 447 genes that were regulated at least 2-fold upon 24 h 4sU treatment (Fig. 2c).

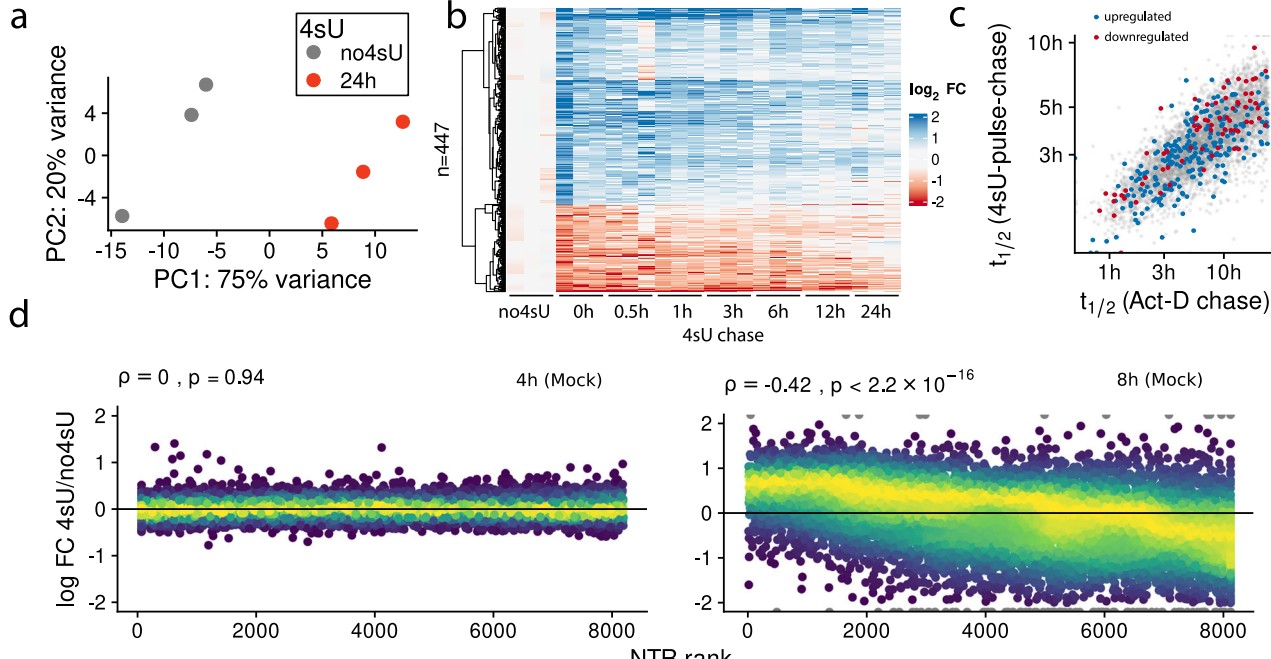

**Fig. 2 | Testing for 4sU toxicity using grandR. a** Principal component analysis of 3 mouse embryonic stem cell samples treated with 100 μM 4sU for 24 h and 3 samples without 4sU treatment (no4sU). The percentages of the explained total variance for both principal components shown are indicated. **b** Heatmap of the 4sU-pulse chase experiment showing log$_2$ fold changes vs the mean of the three 4sU naïve control samples (no4sU) of n = 447 genes that were differentially expressed (>2-fold regulated, 5% FDR, DESeq2 Wald test) between no4sU and the samples after 24 h labeling (0 h 4sU chase). The color scale was chosen to show genes with <1.3-fold regulation in pure white. **c** Scatter plot comparing the RNA half-lives measured using Actinomycin-D (Act-D) chase experiments and using the 4sU-pulse-chase data for n = 5100 genes. The up- and downregulated genes also shown in Fig. 1b are indicated. **d** Scatter plots comparing the ranks of the new-to-total RNA ratios (NTR) of each gene against the log2 fold change of the 4 h (left plot) or 8 h (right plot) sample vs a 4sU naïve sample (untreated mock samples from ref. 18). Rank 0 is the minimal, rank 1 the maximal NTR value. The Spearman correlation coefficients with associated P-values (two-sided approximate t-test) are indicated.

We concluded that the observed expression changes upon 24 h 4sU treatment are due to changes in RNA synthesis, indicating that even for those genes that were regulated >2-fold, half-life estimates from pulse-chase experiments were not biased by long-term 4sU treatment. However, our analyses here demonstrate the importance of assessing potential 4sU induced effects on transcription or RNA stability for pulse-chase designs and long-term labeling in general.

The kinetics of RNA degradation can also be analyzed without chase by monitoring the drop of unlabeled RNA over time. Such a "progressive labeling" design has been used by several studies[1,3,18,22] and provides accurate estimates when timepoints are chosen roughly in the range of the actual RNA half-lives[17,27]. Zuckerman et al.[18] performed siRNA knock-down of the nuclear export factor *NXF1* and used progressive 4sU labeling for 0 h, 2 h, 4 h and 8 h to show that RNA half-lives were not altered. Quality control by grandR revealed that after 8 h, but not before, the gene-wise newly synthesized to total RNA ratio (NTR), which is here equal to the percentage of labeled RNA, was significantly correlated with the $\log_2$ fold change with respect to the (4sU naïve) 0 h time point (Spearman's $\rho = -0.42$, $p < 2.2 \times 10^{-16}$, Fig. 2d). This observed downregulation of short-lived RNAs in the total RNA pool upon 4sU treatment can have technical reasons or can be indicative of a perturbed RNA metabolism in these samples. Importantly for this study the estimated half-lives were not systematically different after excluding the 8 h time point from analysis (Supplementary Fig. 1f). The variance in the differences between the RNA half-lives estimated with and without the 8 h time point, however, was larger for long-lived RNAs, which is not surprising, since accurately estimating long RNA half-lives requires long labeling[17,27]. Thus 4sU concentrations should be chosen such that 4sU induced effects on RNA metabolism are minimized also for later time points. Quality control in grandR can be used to assess such effects and to exclude samples from further analyses.

In summary, grandR facilitates quality control of 4sU labeling experiments and allows to identify problematic genes or samples in the sequencing data by comparing 4sU treated samples with equivalent 4sU naïve controls.

## Comparative analysis of kinetic modeling

The commonly used kinetic model of RNA expression goes back to 1952[28]. In this model, the RNA for a gene is produced with a fixed synthesis rate $\sigma$ and destroyed with a fixed degradation rate $\delta$. Degradation occurs with first-order kinetics, i.e. the number of degradation events per time unit is proportional to the number of RNA molecules. A direct consequence of this model is that with constant rates $\sigma$ and $\delta$, the number of RNA molecules always approaches a steady state level, where the same number of RNA molecules is produced and destroyed per time unit. The speed by which non-steady state levels move towards steady state can be characterized by the RNA half-life $t_{1/2}$, which is the time it takes to move half the way towards steady state. The RNA half-life is independent of the synthesis rate $\sigma$, and, thus, the RNA stability can equivalently be characterized via $\delta$ or $t_{1/2}$.

Different variants of this model have been used to estimate kinetic parameters using nucleotide conversion data with progressive labeling designs: (i) Finkel et al.[22] focused on log transformed estimates of unlabeled RNA and inferred RNA half-lives using simple linear regression (LM). (ii) In Narain et al.[3], we used non-linear least squares regression (NLLS) to fit the full model including $\sigma$ and $\delta$, which has been done in a similar manner in Zuckerman et al.[18]. (iii) Boileau et al.[21] adapted their pulseR package[19] that was originally designed for affinity purified labeled RNA. (iv) Finally, we have presented a Bayesian method to estimate the degradation rate $\delta$ under steady state directly from the NTR[17]. The main difference among methods (i)-(iv) is the error model employed, which might result in differing parameter estimates even though the same kinetic model is used. To compare the accuracy of the four methods, we implemented *in-silico* simulation of nucleotide

conversion RNA-seq experiments in grandR. This uses a previously developed method[29] to simulate read counts. Furthermore, individual 4sU incorporation and conversion events as well as background mismatches as introduced by sequencing errors are simulated for each read, and GRAND-SLAM[17] is used to estimate the NTR for each gene. The simulations are also subject to biological variability (see Methods). All parameters including the read count, overdispersion and half-life distributions as well as the 4sU incorporation and background T-to-C mismatch rates are matched to a recent SLAM-seq data set of SARS-CoV-2 infection[22] as reference. Taken together, we designed the simulation in grandR to mimic real data as closely as possible.

While the estimated RNA half-lives correlated well with the ground-truth for all methods ($R > 0.84$, $p < 2.2 \times 10^{-16}$ for all methods, Pearson correlation; Supplementary Fig. 2), the NLLS method (ii) and the Bayesian NTR method (iv) were significantly more accurate than the LM (i) and pulseR (iii) approaches for data simulated under steady state conditions ($P < 2.2 \times 10^{-16}$ for all comparisons, two-sided Kolmogorov-Smirnov test; Fig. 3a). By contrast, non-steady state conditions resulted in generally more substantial deviations from the ground truth (Fig. 3b). This analysis also shows that the accuracy of the NTR approach (iv) suffers most significantly without steady state. The regression (LM, NLLS) and Bayesian (NTR) approaches report interval estimates. The regression methods had relatively large confidence intervals, indicating that the gaussian noise approximation does not properly model our simulated sequencing data, especially in logarithmic space as done by the LM method (Fig. 3c). The Bayesian NTR method can compute approximate or exact credible intervals. For steady state conditions these exact credible intervals were indeed smaller than the more quickly to compute approximate intervals (inter-quartile ranges: approximate, [0.37–1.23]; exact, [0.30-0.99]; Fig. 3c). Under non-steady state conditions, all deviations were underestimated, most notably for the Bayesian credible intervals, where 88% of the simulated half-lives were outside of the credible interval (Supplementary Fig. 3a). In summary, the non-linear least squares (NLLS) regression (ii) and the Bayesian NTR approach (iv) provided the most accurate estimates under steady state conditions. The NTR method slightly outperformed NLLS, but inherently assumes steady state. We therefore recommend the non-linear least squares regression as the default method for estimating RNA kinetics using progressive metabolic labeling data.

## Choosing number of replicates, time points and sequencing depth

Our simulation also enabled us to assess how many reads, replicates and time points are required to obtain accurate estimates of kinetic parameters. We reasoned that a moderate number between 6 and 12 samples per condition should be used. However, it is a priori unclear whether these should be distributed over many time points, or whether more replication of the same time points is more beneficial. First, we asked whether 6 samples should be used to measure time points 0, 4 and 8 h with two replicates ($2 \times 0 + 4 + 8$ h), or whether single samples at 6 time points ($1 \times 0 + 1 + 2 + 4 + 6 + 8$ h) provide more accurate results. Clearly, the "$1 \times 0 + 1 + 2 + 4 + 6 + 8$ h" design resulted in significantly less variable half-life estimates for genes having RNA half-lives <1 h ($p = 2.8 \times 10^{-6}$, Brown-Forsythe test, Supplementary Fig. 3b) and half-lives between 1 and 2 h ($p = 2.4 \times 10^{-8}$, Brown-Forsythe test). Poorer performance with time points not matching to the true RNA half-lives were indeed not unexpected based on previous theoretical considerations[17,27]. To investigate the effect of missing time points in more detail, we simulated a broad range of potential experimental settings (Fig. 3d). If early (1 h) or late (8 h) time points were missing, the estimates for short-lived or long-lived RNAs, respectively, suffered significantly in general. This became most obvious when we only simulated a single time point (Supplementary Fig. 3c). Thus, to analyze the complete landscape of RNA half-lives multiple time-points

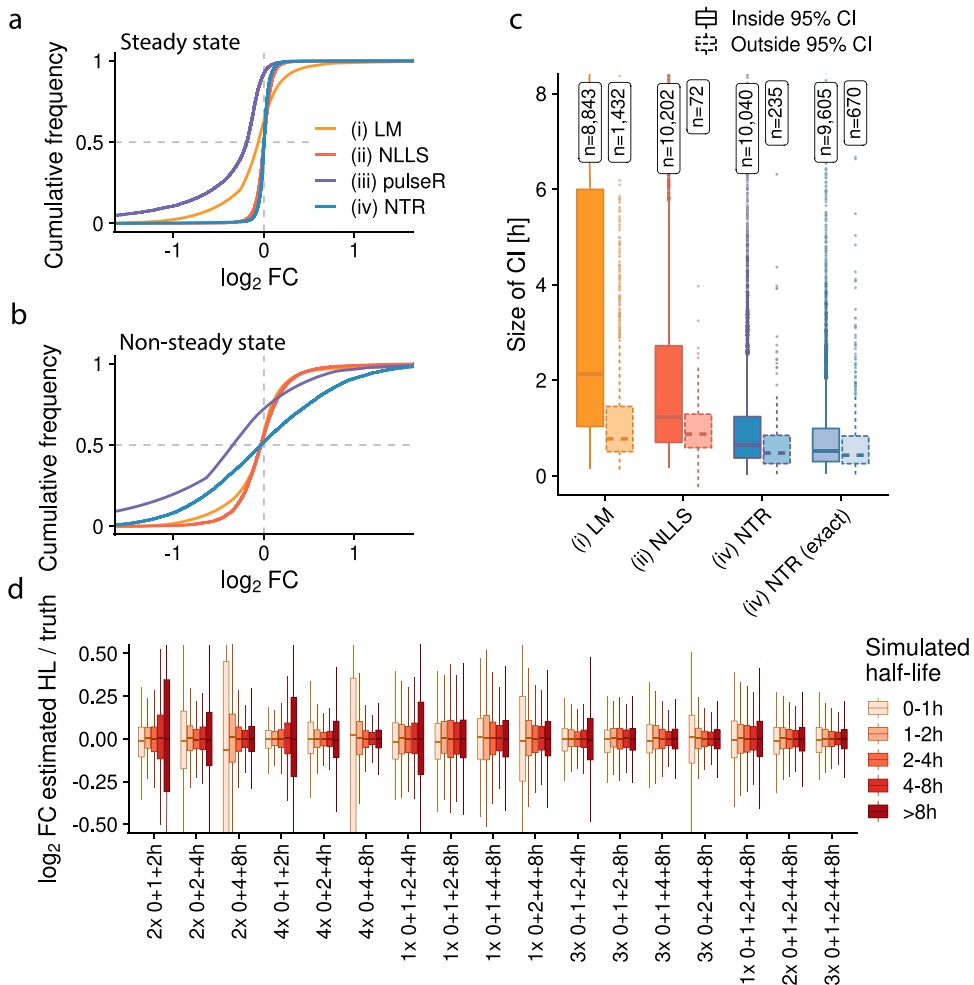

**Fig. 3 | Estimating half-lives using progressive labeling experiments.**
**a**, **b** Empirical cumulative distributions of log2 fold changes of estimated half-lives vs. simulated ground-truth ($n = 10{,}835$ genes) for the linear model (LM), the non-linear least squares method (NLLS), the pulseR method and the Bayesian approach (NTR). In **a**, the ground-truth is simulated under steady state conditions, in **b** the simulation starts from an initial value $a_0 \neq \sigma/\delta$ for each gene (see Methods).
**c** Boxplots showing the sizes of 95% half-life confidence intervals (CI; for LM and NLLS) or 95% half-life credible intervals (for NTR; center line, median; box limits, upper and lower quartiles; whiskers, 1.5x interquartile range; points, outliers). Simulations were performed under steady state conditions. Distributions for genes having the ground-truth inside or outside of the estimated CI are shown separately

and the numbers of these genes are indicated. NTR represents the $\chi^2$ approximation of CIs, NTR (exact) represents exact CIs computed numerically. **d** Boxplots showing log2 fold changes of half-lives estimated by the NLLS method vs the ground truth of simulated data under steady state conditions for genes stratified by RNA half-life (0–1 h, $n = 204$ genes; 1–2 h, $n = 1{,}417$ genes; 2–4 h, $n = 2996$ genes; 4–8 h, $n = 3122$ genes; >8 h, $n = 3096$ genes; center line, median; box limits, upper and lower quartiles; whiskers, 1.5x interquartile range; points, outliers). The distributions for different half-life classes are shown for several experimental settings involving the indicated number of replicates and time points: For example, $2 \times 0 + 1 + 2$ h means two replicates for each of three time points with 0 h, 1 h and 2 h of labeling. 0 h timepoints refer to samples without metabolic RNA labeling.

spanning the whole range of RNA half-lives are required. We next simulated data for a full progressive time course (1 h, 2 h, 4 h and 8 h) with different sequencing depths and numbers of replicates (Supplementary Fig. 3d). Interestingly, increasing the number of replicates per time point boosted the accuracy stronger than increasing the number of reads. This likely is an effect of the biological variability that we simulated. With biological variability the actual RNA half-life is not the same but varies slightly for a gene among all samples (time points and replicates). Additional replicates are thus likely important to accurately include the variability into the kinetic model. Importantly, however, the extent of improvement due to more replicates depends on the magnitude of biological variation. In general, the reported accuracies of the estimates depend on the assumptions made for simulation and the parameters used. We therefore recommend running custom simulations using the functions implemented in grandR to come to an informed decision for planning experiments. In conclusion, our data show that time points must be carefully chosen and

the costs for sample and library preparation must be weighed against the sequencing costs to obtain accurate RNA half-lives.

## Temporal recalibration improves the model fit
4sU is not available for transcription immediately once the cells are cultured on 4sU media, but first crosses cell membranes and is then processed by the pyrimidine salvage pathway before it is available as substrate for transcription (reviewed in ref. [5]). Thus, the concentration of active 4sU increases until saturation, and RNA that was transcribed significantly before reaching saturation contains fewer 4sU than RNA transcribed later. Such an increase in the 4sU concentration has been described recently[30]. Therefore, especially for earlier time points, the effective labeling time is expected to be much shorter than the nominal labeling time. For example, the effective labeling time of a nominally 1 h sample might only by 40 min, since during the first 20 min, the concentration of activated 4sU has been too low to induce many 4sU incorporation events.

To test this effect, we used grandR to estimate RNA half-lives for published data of Calu-3 cells infected with SARS-CoV-2 and mock infected control cells[22]. Indeed, the residuals of the model fit were mostly negative for the 1 h time points, and more balanced at later time points (Fig. 4a, b). The effective labeling time for a sample is a global parameter that is common to all genes. Moreover, the temporal behavior of new and old RNA for all genes is constrained by the kinetic model: The new RNA levels from different time points lie on a single curve (the model fit) that approaches a specific steady state level and only has two degrees of freedom. Conceptually, if the effective labeling time for a 1 h sample actually is 40 min, the new RNA level for this sample for each gene would be below the model fit, and moving this sample to the 40 min time point would correct for that (Fig. 4a). Thus,

we reasoned that it should be possible to estimate effective labeling times by maximizing the joint likelihood of all gene specific synthesis and degradation rates and the effective labeling times.

We first tested this temporal recalibration by simulated time courses where we artificially reduced the nominal labeling times by 20–40 min. The recalibrated labeling times were on average within 1.03-fold of the true effective labeling time (Supplementary Fig. 4a), and estimation of kinetic parameters was completely rescued after recalibration (Supplementary Fig. 4b). To further test our recalibration method, we compared RNA half-lives in mESCs that were estimated (i) by Herzog et al.[7] using pulse-chase SLAM-seq data, (ii) by Herzog et al. using Actinomycin-D (Act-D) chase RNA-seq data, (iii) by grandR using the same pulse-chase data, (iv) by grandR using progressive labeling

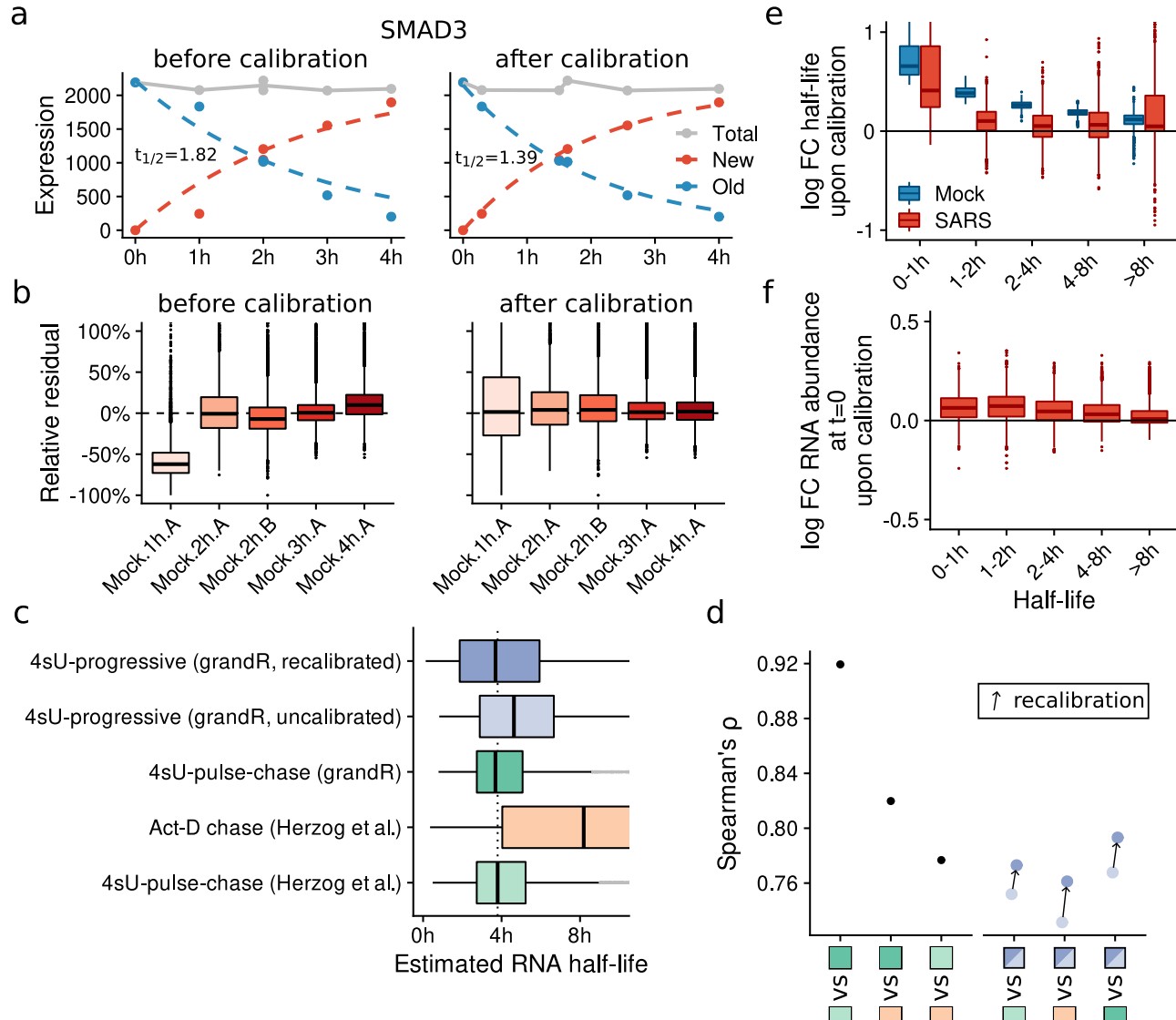

**Fig. 4 | Temporal recalibration of SARS-CoV-2 SLAM-seq data. a** Progressive labeling plots of the SMAD3 gene before (left) and after (right) temporal recalibration. Points represent the total, new or old read count of SMAD3 at the indicated time after labeling. Dashed lines show the model fit (NLLS method). Estimated half-lives are indicated. **b** Boxplots showing relative residuals from the model fit (NLLS method) before (left) and after (right) temporal recalibration of n = 9162 genes for all samples (center line, median; box limits, upper and lower quartiles; whiskers, 1.5x interquartile range; points, outliers). **c** Boxplots showing the distribution of RNA half-lives estimated from different data sources and by different methods as indicated (n = 6148 genes; center line, median; box limits, upper and lower quartiles; whiskers, 1.5x interquartile range). The dotted line

indicates the median of the pulse-chase experiment as estimated by Herzog et al. For clarity, the x-axis is cut at 10 h. **d** Spearman correlation coefficients shown for all three pairwise comparisons of the reference RNA half-lives, and showing the improvement due to recalibration for all three references. Color codes indicating the comparisons are the same as in **c**. **e, f** Boxplots showing log2 fold changes (recalibrated vs. uncalibrated) of half-lives for the mock and virus infected (SARS) samples (**e**) or of the estimated initial abundances $a_0$ (recalibrated vs. uncalibrated) for the non-steady state infected samples (**f**) for n = 9,163 genes. Separate distributions for genes from different half-life classes are shown (center line, median; box limits, upper and lower quartiles; whiskers, 1.5x interquartile range; points, outliers).

data without or (v) with recalibration (Fig. 4c, d). The two half-life estimates from the pulse-chase data were in excellent agreement, both with respect to correlation (Spearman's $\rho = 0.92$) and magnitude (Herzog et al.: median = 3.80 h, grandR, median: 3.69 h). Interestingly, the Act-D data were well correlated as well ($\rho = 0.78$ vs pulse-chase estimated by Herzog et al., $\rho = 0.82$ vs pulse-chase estimated by grandR), as described[7], but showed substantially longer half-lives (median: 8.19 h). With respect to all three references (i–iii), recalibration slightly improved the correlation of half-lives estimated from progressive labeling (difference of $\rho > 0.02$ for all references, Fig. 4d). Considering the noisy estimates of RNA half-lives, also these modest improvements are remarkable, and statistically highly significant ($p < 2.2 \times 10^{-16}$ for all three references, $t$-test). More importantly, recalibration rectified the magnitude of the estimated half-lives (uncalibrated: median = 4.63 h; recalibrated: median=3.69 h, Fig. 4c).

We then recalibrated the labeling times for the SARS-CoV-2 data. Indeed, the residuals became smaller for all samples and were now symmetric (Fig. 4a, b). Globally, temporal recalibration affected short half-lives stronger than long half-lives (Fig. 4e), presumably because early time points are the most informative to estimate short half-lives. Moreover, for the virus infected samples there were substantially more gene-specific differences at the 1 h time point, indicating that without steady state assumption, the first time point is important to estimate the initial abundance $a_0$. Indeed, the estimates of $a_0$ exhibited the same gene specific differences like half-life estimates upon calibration (Fig. 4f). In conclusion, due to the kinetics of 4sU uptake and activation, the effective labeling time might differ from the nominal labeling time, especially for short labeling. For progressive labeling experiments, this can be corrected by temporal recalibration.

## Estimating changes in synthesis or degradation from snapshot experiments

Nucleotide conversion RNA-seq has also applications beyond progressive labeling time courses. We and others showed that new RNA from a single "snapshot" timepoint is more sensitive to detect short-term changes of gene expression than standard RNA-seq without metabolic RNA labeling, e.g. upon virus infection[12], drug treatment, or acute depletion of transcription factors via degron systems[3,16]. So far, analyses of snapshot samples have been performed in an ad-hoc manner by the application of standard differential gene expression tools on estimated new or old RNA.

We have previously shown that steady state half-lives can, in principle, be estimated from a single snapshot sample based on the NTR[17]. Here we show that both synthesis and degradation rates ($\sigma$ and $\delta$) can be estimated when not in steady state as long as a "reference" sample is available from the time point where labeling was initiated, or from a prior time point (see Methods).

Such estimates of $\sigma$ and $\delta$ might be highly inaccurate due to the NTR quantification uncertainty, due to a labeling time not matching the gene specific RNA half-life, or sampling noise due to low numbers of reads. In addition to these technical factors, $\sigma$ and $\delta$ are also subject to biological variability among replicate samples. To control these factors, we developed a Bayesian hierarchical model to estimate the joint posterior distribution of $\sigma$ and $\delta$ as well as the joint posterior of $\log_2 \frac{\sigma_A}{\sigma_B}$ and $\log_2 \frac{\delta_A}{\delta_B}$ for differential analysis of two samples A and B.

To test our approach, we simulated data for two conditions with 2 h labeling. We left one condition at steady state, for the other we either perturbed synthesis or degradation rates, or left them unperturbed as control. The maximum-a-posteriori log fold change estimates of both $\sigma$ and $\delta$ were unbiased, and much more accurate for $\sigma$ (root mean square deviation (RMSD) = 0.047; Fig. 5a) than for $\delta$ (RMSD = 0.513; Fig. 5b). Estimated changes in $\sigma$ reflected the true change in synthesis rates slightly more accurately than new RNA (RMSD = 0.062; Fig. 5c). Counterintuitively, the old RNA fold change did not correspond to the true fold change of RNA half-lives (Fig. 5d).

Indeed, our model shows that the log fold change of old RNA does not correspond to a relative change in degradation rates or RNA half-lives, but there is a more complex relationship between the observed fold change and the change in RNA stability (see Methods). Previously, an observed new RNA fold change has been equated with a change in synthesis rate[15]. However, our model shows that new RNA fold changes are also affected by changes in degradation rates, predominantly for genes with short-lived RNAs. Indeed, we observed significant changes in new RNA when only the degradation but not the synthesis rates were changed, which was restricted to genes with short-lived RNAs (Fig. 5e). Thus, if synthesis rates are changed, these changes are directly reflected on the change in new RNA. However, the converse is not true, since a change in RNA levels can also be due to a change in RNA stability especially for short-lived RNAs. Of note the estimated synthesis rate changes by our Bayesian model were not affected by changes of RNA stability. For unperturbed controls, estimated changes of $\delta$ exhibited more variance than estimated changes of $\sigma$. This effect was much less pronounced for genes with short RNA half-lives, or when the labeling duration was 4 h instead of 2 h (Fig. 5f).

It is important to note that the kinetic model assumes constant rates of RNA synthesis and degradation during the time of labeling. If this assumption is not met, the estimated rates represent weighted averages of these varying rates within the labeling time (Supplementary Note 1). To test this, we again simulated data leaving one condition at steady state (and with constant rates) as control and let either the synthesis or degradation rate slowly approach a perturbed state over the time of labeling instead of setting them to a new value at the onset of labeling. The log fold changes vs control for the synthesis rates again reflected the true changes more accurately (RMSD = 0.087, Fig. 5g) than for the degradation rates (RMSD = 1.609, Fig. 5h). For both, the estimated log fold changes had a on average 80% lower magnitude than the true final rates after 2 h of labeling corresponding to averaging over time for the estimated rates. We concluded that time-varying synthesis and degradation rates can be estimated using grandR, and that estimates correspond to weighted averages over the labeling time.

We then applied our Bayesian model for changes of RNA stability to the 2 h time point of the SARS-CoV-2 data[22], revealing that the degradation rate changes recapitulated the changes identified by modeling the full progressive labeling time course ($R = 0.7$, $p < 2.2 \times 10^{-16}$, Pearson correlation; Fig. 5i).

In summary, in contrast to previously used fold changes of old and new RNA, the maximum-a-posteriori estimates of our hierarchical model provide directly interpretable log fold changes of synthesis and RNA half-lives from snapshot data.

## ROPE analysis of significant changes of $\sigma$ and $\delta$

We analyzed "regions of practical equivalence" (ROPE)[31] to quantify significant changes of synthesis or degradation using our Bayesian approach. As a measure of significance, we used the posterior probabilities $P_\sigma$ or $P_\delta$ of the $\log_2$ fold change (synthesis or degradation, respectively) being either $< -0.25$ or $> 0.25$. As a comparison, we analyzed Benjamini-Hochberg adjusted $P$-values $q_{new}$ and $q_{old}$ computed by DESeq2[23] for new and old RNA, respectively.

We first analyzed our simulated data (2 h labeling) where RNA synthesis rates were perturbed. As expected from overall $n = 10,835$ genes, virtually none had $P_\delta > 0.9$ ($n = 127$, 1.2%) or $q_{old} < 0.01$ ($n = 0$) independent of the true change of RNA synthesis (Fig. 6a). Notably, of the $n = 3388$ genes simulated to be more than 2-fold up- or down-regulated, $n = 3249$ (95.9%) and $n = 3174$ (93.7%) genes had $P_\sigma > 0.9$ and $q_n < 0.01$, respectively (Fig. 6a). Thus, ROPE analysis of our Bayesian model and DESeq2 analysis of new and old RNA showed similar sensitivity and specificity when only RNA synthesis rates are changed.

Next, we focused on simulated data where RNA half-lives were perturbed. From overall $n = 10,835$ genes, $n = 40$ (0.4%) had $P_\sigma > 0.9$

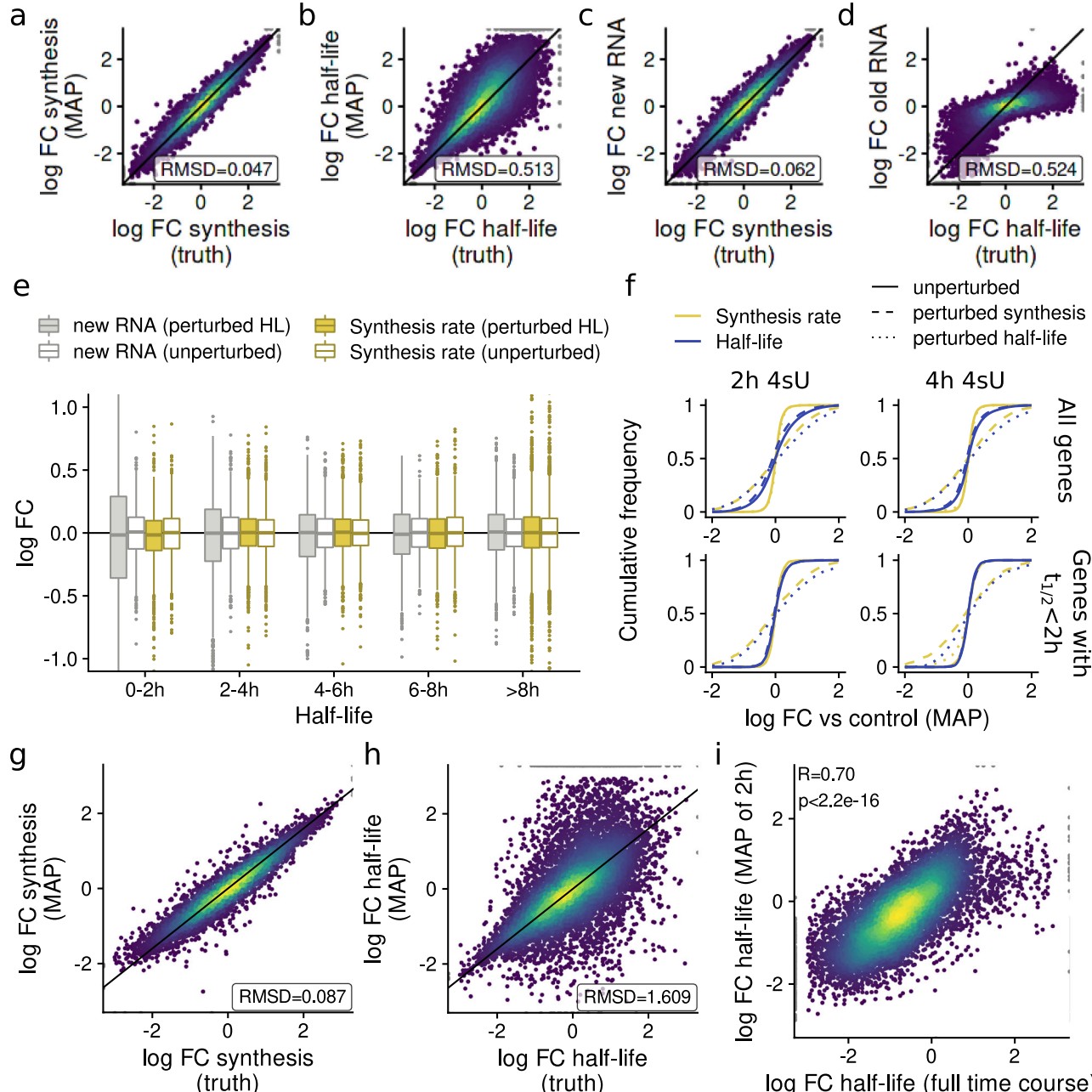

**Fig. 5 | Estimating changes in synthesis or degradation from snapshot experiments. a–d** Scatterplots comparing simulated log2 fold changes against maximum-a-posteriori (MAP) estimates of RNA synthesis log2 fold changes (**a**), MAP estimates of RNA half-life log2 fold changes (**b**), observed new RNA log2 fold changes (**c**) or old RNA log2 fold changes (**d**). Three replicates at 2 h of labeling were simulated after perturbing synthesis (**a**, **c**) or half-lives (**b**, **d**) for 2 h and compared against unperturbed controls. The root mean square deviations (RMSD) over all $n = 10,835$ simulated genes are indicated for each comparison. **e** Boxplots showing the log2 fold changes of new RNA or of estimated synthesis rates either for the simulated samples with perturbed RNA half-lives (perturbed HL) or the unperturbed samples vs the controls. Separate distributions for genes from different simulated half-live classes are shown as indicated (0–2 h, $n = 1621$ genes; 2–4 h, $n = 2996$ genes; 4–6 h, $n = 1940$ genes; 6–8 h, $n = 1182$ genes; >8 h, $n = 3096$ genes; center line, median; box limits, upper and lower quartiles; whiskers, 1.5x inter-quartile range; points, outliers). **f** Empirical cumulative distributions showing log2

fold changes of either estimated synthesis rates (yellow) or RNA half-lives (blue). For each distribution, either unperturbed samples (solid lines), samples with perturbed synthesis rates (dashed lines) or samples with perturbed half-lives (dotted lines) were compared against controls. Distributions are shown for all genes, only for genes with short RNA half-lives $t_{1/2} < 2$ h, and for simulated labeling of 2 h or 4 h, as indicated. **g, h** Scatterplots comparing simulated log2 fold changes against MAP estimates of RNA synthesis log2 fold changes (**g**) or RNA half-life fold changes (**h**). Three replicates at 2 h of labeling were simulated with synthesis rates (**g**) or degradation rates (**h**) slowly approaching a perturbed state during 2 h of labeling. RMSDs over all $n = 10,835$ simulated genes are indicated for each comparison. **i** Scatterplot comparing log2 fold changes of RNA half-lives estimated from the full progressive labeling time courses using the NLLS method (x-axis) or the MAP estimator from our Bayesian model using the 2 h time point only. The Pearson correlation and the associated *P*-value (two-sided *t*-test) are indicated.

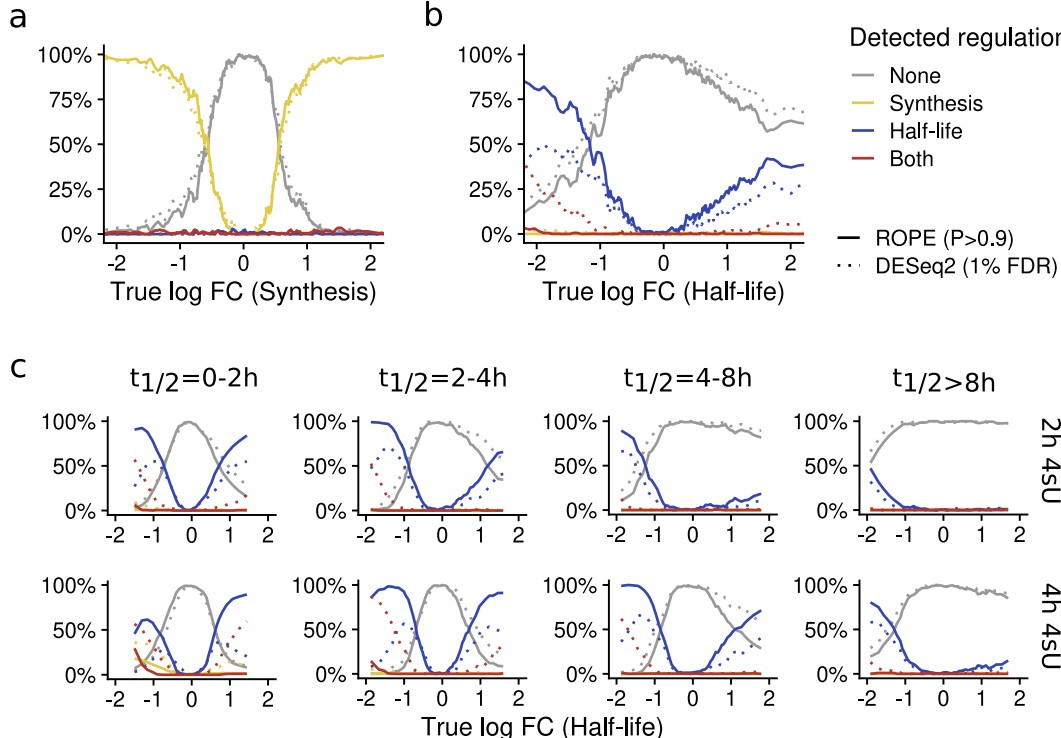

**Fig. 6 | Region of practical equivalence analysis of simulated data.** Line plots comparing two criteria for differential regulation of synthesis rates (**a**) or half-lives (**b, c**) are shown. For **a**, the two criteria are the ROPE probability for synthesis $P_\sigma > 0.9$ and the DESeq2 *P*-value (two-sided Wald test, Benjamini-Hochberg multiple testing adjusted) for new RNA $q_n < 0.01$. For **b** and **c**, the two criteria are the ROPE probability for degradation $P_\delta > 0.9$ and the DESeq2 *P*-value (two-sided Wald test, Benjamini-Hochberg multiple testing adjusted) for old RNA $q_o < 0.01$. The x axis represents rolling statistics (bin width 200 genes) over the log2 fold change of synthesis rates (**a**) or RNA half-lives (**b**) for the simulations with perturbed synthesis and half-lives vs. control, respectively. The different lines show the percentage of genes in a bin with detected regulation in synthesis, half-life, both or none. **a** and **b** show all genes for 2 h of 4sU labeling. **c** shows genes of different half-life classes and for 2 h or 4 h of 4sU labeling, as indicated.

and $n = 407$ (4.0%) had $q_{new} < 0.01$ (Fig. 6b). These hundreds of genes with significant changes in new RNA predominantly had down-regulated RNA half-lives ($n = 302$ out of overall 1641 genes with >2-fold downregulated RNA half-lives, 18.4%). Thus, as shown above, new RNA for some genes exhibited significant changes when only RNA half-lives are changed. By contrast, our Bayesian approach can accurately differentiate between changes in synthesis and degradation. Moreover, out of $n = 3390$ genes with >2-fold regulated half-lives, $n = 1692$ (49.9%) had $P_\delta > 0.9$ and $n = 1351$ (39.8%) had $q_{old} < 0.01$ (Fig. 6b), indicating that our Bayesian approach is also more sensitive than analyzing old RNA for detecting changes in RNA stability.

Interestingly, the sensitivities of our Bayesian approach for detecting changes in RNA stability were asymmetric (65.7% for down-regulated RNA half-lives and 33.0% for upregulated RNA half-lives; Fig. 6b). To investigate this further, we stratified genes according to their unperturbed RNA half-lives. For genes with RNA half-lives of <2 h, sensitivities indeed were symmetric, but exhibited increasing asymmetry for longer half-lives (Fig. 6c). This asymmetry can be explained by the fact that half-lives not matching to the duration of 4sU labeling cannot be estimated accurately. To corroborate this, we repeated these analyses with 4 h of simulated 4sU labeling, which resulted in symmetric sensitivities for average half-lives of 2-4 h. In summary, our Bayesian approach can accurately differentiate between effects on RNA synthesis and degradation.

**Bayesian analysis indicates target gene specific differences of regulation by acute *BANP* depletion**

We utilized our Bayesian modeling approach for the analysis of published data from cells after degron-mediated depletion of *BANP*, which has recently been revealed to bind to unmethylated CGCG motifs in

CpG islands to promote transcription of a set of essential genes[16]. For this study, samples from multiple timepoints (1 h, 2 h, 4 h, 6 h and 20 h) after depletion of *BANP* were labeled with 4sU prior to sequencing. Importantly, the samples from the 4 h timepoint and later were labeled for 2 h, but shorter labeling of 30 and 90 min was applied for the 1 h and 2 h timepoints, respectively. Due to these different labeling times, new RNA is not directly comparable among the samples and inference of $\sigma$ is required to interpret the data. We first recalibrated labeling times. Of note, the recalibration approach described above requires a progressive labeling design, and, therefore, cannot be used here. Thus, we developed a second recalibration method: We assume that, globally on average, degradation rates are not changed due to *BANP* depletion across the timepoints. Thus, we adapted the effective labeling time such that the median log fold change of the half-lives resulting from the adapted labeling time vs the half-lives from the control samples without *BANP* depletion was 0 for each sample. Indeed, after this recalibration by matching the half-life distribution to this reference, the distribution for RNA half-lives estimated for each timepoint were largely indistinguishable (Supplementary Fig. 5a), and the same was also true for RNA synthesis rates (Supplementary Fig. 5b).

For each timepoint, the RNA synthesis log fold changes for *BANP* targets determined by ChIP-seq[16] were significantly and consistently shifted towards negative values compared to non-targets (Fig. 7a). This is remarkable especially for the 1 h timepoint, where 4sU labeling only was 30 min, and suggests that synthesis rates were reduced immediately once *BANP* was depleted from cells, and then stayed constant for at least 20 h. To further investigate this, we analyzed the synthesis log fold change posterior distributions for individual genes. This revealed that there were substantial gene specific differences, with some *BANP* targets like *Taf1d* showing gradually decreasing synthesis rates with

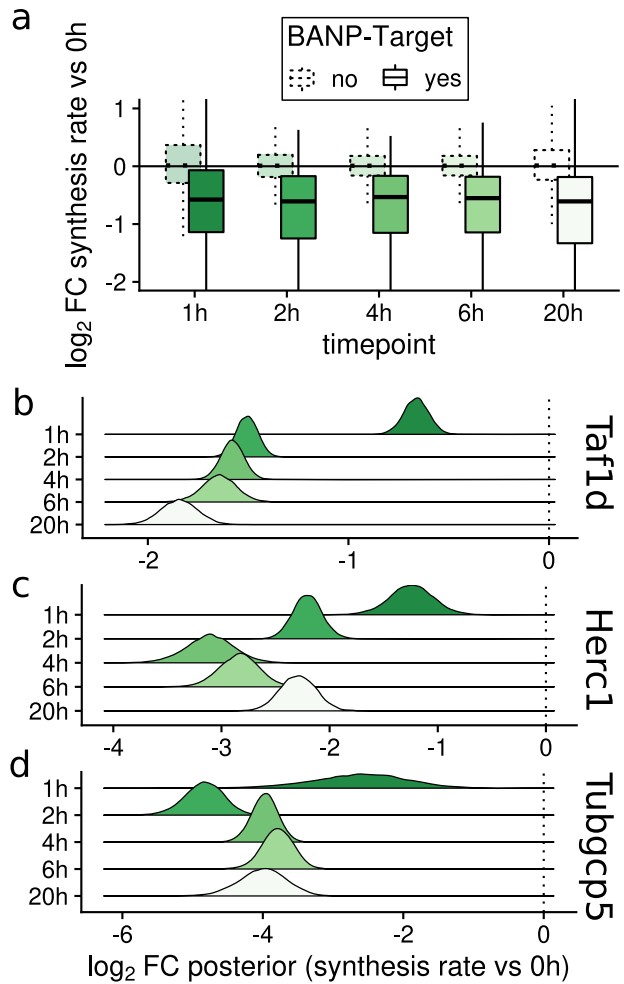

**Fig. 7 | Dynamic regulation of synthesis rates upon acute *BANP* depletion.**
**a** Boxplots showing log2 fold changes of synthesis rates for several experimental
time points vs the 0 h time point. Distributions of *BANP* target genes (*n* = 455 genes,
according to ChIP-seq experiments) are shown separately from non-target genes
(*n* = 10,641 genes; center line, median; box limits, upper and lower quartiles;
whiskers, 1.5x interquartile range; points, outliers). **b–d** Estimated posterior den-
sities for Taf1d (**b**), Herc1 (**c**) and Tubgcp5 (**d**) of log2 fold changes of the synthesis
rates of the indicated time points vs. the 0 h time point.

The error models used by previous packages are not designed to
handle the enormous bias of underestimating labeled RNA by counting
reads with mismatches. The pulseR package, as a notable exception,
has been adapted to include an additional nuisance parameter for
this[21]. In theory, and as empirically shown by our simulation study,
estimated parameters are more accurate when (i) bias due to limited
labeling (the LM, NLLS and NTR models) and (ii) the inherent quanti-
fication uncertainty (the NTR model) are directly taken into account.
Another disadvantage of adapting methods that have been developed
for the affinity purification approach is that they are agnostic of the
highly variable number of uridines across the reads, which impacts on
the likelihood of observing one of the infrequent 4sU incorporation
events. Notably, this is a different, and more severe, issue than the
differences in uridine content among full length RNAs ("uridine bias")
that is relevant to purification approaches and taken into account by
some of the previous tools[19].

It is well described in the literature that nucleoside concentrations
must be optimized for specific cell types and desired labeling
times[6,7,26]. Toxicity of too highly concentrated 4sU has previously been
assessed by testing for cell viability. However, here we show that
expression estimates can be affected before cell viability suffers. We
therefore advocate that all studies employing metabolic RNA labeling
must report the extent of any effect of 4sU on expression. Since levels
of short-lived RNAs quickly decline when transcription is globally
inhibited, a correlation of the fold changes for 4sU treated samples vs.
corresponding 4sU naïve controls with the NTRs can be used as sur-
rogate marker for transcriptional defects due to 4sU treatment, as
implemented in grandR. However, it is important to note that also
technical effects such as decreased efficiency of reverse transcription
or less accurate mappability of reads with many mismatches can result
in such a correlation.

RNA degradation rates have previously been estimated using
progressive labeling time courses using different computational
methods. All these approaches employed the kinetic model described
in the Methods section but they differ in their choice of the error
model. Due to noise introduced by the inference of the NTR, the actual
errors of normalized new and old RNA likely are differently distributed
than standard RNA-seq data. Our simulations indicate that the errors
are well approximated by a gaussian distribution. We recommend
using the non-linear least squares fitting procedure as the general tool
for fitting the kinetics of RNA expression. The Bayesian approach
provides slightly more accurate results and better error bounds but
can only be used under steady state conditions.

A major caveat of metabolic RNA labeling experiments is that the
effective labeling time might not correspond to the nominal labeling
time. grandR provides tools to test for this critical issue and recalibrate
labeling times: For progressive labeling time courses, asymmetric
residuals of early time points indicate shorter effective labeling times.
Using the labeling times as additional independent variables when
jointly fitting the kinetics for all genes, as implemented in grandR, can
be used to estimate the effective labeling times. For snapshot experi-
ments, labeling times can be recalibrated based on additional
assumptions, e.g. based on reference RNA half-lives that must be known
a-priori. For the *BANP* data set, we made the assumption that globally,
RNA half-lives are not affected by acute depletion of *BANP*, and there-
fore used the estimated half-lives of the untreated control sample as a
reference. Testing for effective labeling is critical when samples with
distinct labeling times are compared, and for estimating RNA degra-
dation and, to a lesser extent, synthesis rates in absolute terms.

A limitation of grandR is that it currently does not implement
methods to estimate kinetic parameters beyond RNA synthesis and
half-lives. RNA processing could in principle be analyzed using reads
mapped to intronic regions, as it is implemented in the DRiLL and
INSPEcT packages[4,20]. However, the amount of intronic RNA among
total RNA is relatively low in comparison to 4sU purified RNA after a

efficient downregulation only later than the 1 h timepoint (Fig. 7b) and
for others like *Herc1* (Fig. 7c) or *Tupgcp5* (Fig. 7d) synthesis rates
dropped early and rose again later suggesting negative feedback
loops. In conclusion, the Bayesian hierarchical model implemented in
grandR can be used to uncover detailed information about gene reg-
ulation from snapshot experiments reflecting genome-wide trends and
to generate testable hypotheses of individual genes.

## Discussion

Nucleotide conversion RNA-seq is now widely used to infer kinetic
parameters of RNA expression but there is still a lack of computational
tools for data analysis. Our goal for developing grandR was to provide
a comprehensive and easy-to-use toolkit to facilitate a broad range of
different analysis steps for such data.

All so far available packages for analyzing experiments involving
metabolic RNA labeling have been designed to work with two (or
three) fractions per sample, one sequenced library of purified labeled
RNA, and one of total (and/or unlabeled) RNA. In principle, these
packages can also be used for nucleotide conversion data after split-
ting labeled and unlabeled RNA by extracting reads containing T-to-C
mismatches. This bioinformatic separation comes with disadvantages:

short labeling pulse. Thus, to obtain accurate parameters of RNA processing, we recommend affinity purification and the use of either DRiLL or INSPEcT. More work is required to explore how intronic reads in nucleotide conversion RNA-seq can be used to infer parameters about RNA processing. Another limitation is that the kinetic model employed by all methods in grandR inherently assumes constant rates of synthesis and degradation during the time of labeling. This might not be true in some scenarios. For example, the reduced RNA half-lives observed for SARS-CoV-2 likely result from the general host shutoff protein nsp1 encoded by SARS-CoV-2[32]. It is therefore very likely that degradation of cellular mRNAs depends on the abundance of nsp1, which increases substantially over the first few hours of infection. Thus, the degradation rate at 4 h post infection (corresponding to the 1 h labeling time point in the SLAM-seq data from ref. 22) likely is different at 7 h post infection (the final 4 h labeling time point), and the estimated RNA half-lives estimated across the progressive labeling time course represent averages. When multiple snapshots over time have been measured, grandR can estimate rates for each snapshot. In such a setting, rates are only assumed to be constant within the time in between each snapshot and its reference (usually the previous snapshot). Thus, dynamically changing rates over time can be detected by analyzing their posterior distributions using the Bayesian model implemented in grandR (see Fig. 7b–d). This approach approximates dynamically changing rate parameters using a piecewise constant function. grandR currently does not support more sophisticated approximations such as the piecewise linear functions that are implemented in INSPEcT[20] to analyze non-constant rates. Of note, for both abrupt changes of rates affecting a single time point and slowly changing rates, the class of approximating function is of low importance.

While our manuscript was under review, the bakR package was published[33]. This package implements a statistical test to identify genes with significantly different NTRs using a Bayesian hierarchical model. This test extends our previously published Bayesian approach of transforming NTRs into RNA half-lives (here called model (iv))[17]. Due to this one-to-one correspondence, the test in bakR can be used to identify significant differences in RNA half-lives, importantly, however, only under steady state conditions. bakR does not attempt to model non-steady state conditions, which is an inherent part of the Bayesian hierarchical model implemented in grandR.

Nucleotide conversion approaches greatly reduced the burden on the wet-lab side for conducting metabolic RNA labeling experiments but introduced the need for more sophisticated tools for their computational analysis. Complementing our GRAND-SLAM software for primary processing of such data, we developed the grandR package as a general toolkit to aid researchers to further analyze and interpret such data. Here, we demonstrated that additional quality control measures are necessary for such data to exclude effects of 4sU on transcription, and that short labeling times often require recalibration. For both methods, grandR provides high-level functions. Furthermore, grandR enables researchers to estimate synthesis and degradation rates for both progressive labeling as well as snapshot experiments without requiring steady state assumptions. Finally, grandR provides a web-based interface for exploratory data analysis.

## Methods

### SLAM-seq preprocessing
All SLAM-seq data used here were processed using the GRAND-SLAM pipeline[17]. Fastq files were downloaded from the SRA database. The accession number were: GSE99970 for the pulse-chase 4sU labeling data set in mESCs from ref. 7, GSE99974 (only wt cells) and GSE99972 for the progressive labeling in mESCs[7], GSE139151 for the NXF1 knockdown data set from ref. 18, GSE162323 for the SARS-CoV-2 data set from ref. 22, and GSE155604 for the BANP depletion data set from ref. 16. Adapter sequences were trimmed using cutadapt (version 3.4)

using parameters "-a AGATCGGAAGAGCACACGTCTGAACTCCAGTCA -A AGATCGGAAGAGCGTCGTGTAGGGAAAGAGTGT" for the SARS-CoV-2 data. For the other data sets, reads were already pre-trimmed on SRA. Then, bowtie2 (version 2.3.0) was used to map read against an rRNA (NR_046233.2 for mESCs and BANP, and U13369.1 for NXF1 and SARS-CoV-2) and Mycoplasma database using default parameters. Remaining reads were mapped against target databases using STAR (version 2.5.3a) using parameters "–outFilterMismatchNmax 20–outFilterScoreMinOverLread 0.4–outFilterMatchNminOverLread 0.4–alignEndsType Extend5pOfReads12–outSAMattributes nM MD NH". We used the murine genome for mESCs and BANP, the human genome for NXF1, and the combined human and SARS-CoV-2 (NC_045512) genome for SARS-CoV-2. All genome sequences were taken from the Ensembl database (version 90). Bam files for each data set were merged and converted into a CIT file using the GEDI toolkit[34] and then processed using GRAND-SLAM (version 2.0.7; for the NXF1 data set, GRAND-SLAM 2.0.5 g was used) with parameters "-trim5p 15 -modelall" to generate read counts and NTR values on the gene level, taking into account all reads that are compatible with at least one isoform of a gene. For all mESC data sets, we performed a second GRAND-SLAM run utilizing the RNA 3′ end annotation provided in ref. 7 and setting the parameters -conv 0.025 -err 0.00035 (which are the global parameters for the mixture model that were estimated from the 24 h samples alone) since GRAND-SLAM did not estimate these parameters for several of the individual samples (e.g. 45 min pulse, 12 h chase) due to extremely low amounts of labelled RNA. These runs utilizing the custom 3′ end annotation were used for the the analyses shown in Fig. 2b, c and Supplementary Fig. 1b-d and the comparative analyses in Fig. 4c, d. For the Actinomycin-D derived half-lives, we utilized the processed data file provided on GEO with accession GSE99975.

### Statistical analyses
All correlation coefficients were computed using the cor function of R (version 4.2.1). Wald tests for differential expression and principal analyses were computed using DESeq2 with default parameters (version 1.36) using the PairwiseDESeq2 or PlotPCA functions of grandR, respectively. The t-test for the differences of correlation coefficients was computed using the psych package (version 2.2.9).

### Read simulation
To simulate a nucleotide conversion RNA-seq experiment for n genes with relative abundances $a_1, \ldots, a_n$, $\sum_i a_i = 1$, overdispersion parameters $d_i$ and total read count $N$, first n random numbers $C_i$ are drawn from negative binomial distributions $NegBinom(a_i N, d_i)$. We use the parametrization $NegBinom(\mu, d)$ such that the mean is $\mu$ and the variance is $\mu + d\mu^2$.

To simulate the "measured" NTR for gene i given the true NTR $ntr_i$, we sampled the number of uridines $u_r$ covered by each of the $c_i$ reads from a binomial distribution $Binom(rl, p_u)$, where $rl$ is the user-defined read length (used here: 75), and the probability for an uridine at any position $p_u$ is sampled from a beta distribution with user-defined average uridine content (used here: 0.25) and standard deviation thereof (used here: 0.05). For each read r, then the number of conversions $tc_r$ is sampled from a binomial mixture distribution $BinomMix(u_r, p_e, p_c, ntr_i)$ defined by the probability function

$$P(k; u_r, p_e, p_c, ntr_i) = (1 - ntr_i)B(k; u_r, p_e) + ntr_i B(k; u_r, p_c). \quad (1)$$

Here, $B(k; n, p) = \binom{n}{k} p^k (1-p)^{n-k}$ is the probability function of the binomial distribution. We used a sequencing error rate of $p_e = 10^{-4}$ and a T-to-C conversion rate of $p_c = 0.04$. Of note, $D = (u_r, tc_r)$, $r \in \{1, \ldots, c_i\}$ for a gene i represent the sufficient statistics for the

GRAND-SLAM model. Then, GRAND-SLAM is used to obtain the maximum-a-posteriori (MAP) estimate for the NTR by numerically maximizing the binomial mixture log likelihood function[17], i.e. we used a uniform prior. In addition, it also computes the Beta approximation of the posterior distribution of $ntr_i$ as described[17].

This procedure is implemented in the function *SimulateReadsForSample* of grandR and simulates the read count $C_i$ and the NTR $\Xi_i$ for $n$ genes based in several user-defined parameters as described above. grandR also provides the higher-level function *SimulateTimeCourse*. Based on a time point $t$ RNA synthesis rates $\sigma_i$, degradation rates $\delta_i$, initial abundances $a_{0i}$ and global synthesis and degradation variance parameters $v_\sigma$ and $v_\delta$ (here both were set to 1.05), this function computes both the relative abundances $a_i$ and the new-to-total RNA ratio $ntr_i$, i.e. the parameters to *SimulateReadsForSample* as follows: To model biological variability, we define $\tilde{\sigma}_i = \sigma_i \cdot 2^\epsilon$ where $\epsilon$ is gaussian noise. Here, the noise level was chosen such that the 95% quantile of the gaussian is equal to $\log_2 v_\sigma$. This way, 90% of all $\tilde{\sigma}_i$ are expected to be at most $v_\sigma$-fold less or greater than $\sigma_i$. Equivalently, we defined $\tilde{\delta}_i$. Then, the abundance of new and old RNA at time $t$ is computed as $a_i^{(new)}(t) = \frac{\tilde{\sigma}_i}{\tilde{\delta}_i}(1 - e^{-t\tilde{\delta}_i})$ and $a_i^{(old)}(t) = a_{0i}e^{-t\tilde{\delta}_i}$, respectively. Thus, $a_i = a_i^{(new)}(t) + a_i^{(old)}(t)$ and $ntr_i = a_i^{(new)}(t)/a_i$.

For all simulations here, we used the "mock" samples from the SARS-CoV-2 data set to compute the relative abundances $a_i$ and estimated the overdispersion parameters $d_i$ using the function *estimateDispersions* from the DESeq2 package. The reference synthesis rates $\sigma_i$ and degradation rates $\delta_i$ were estimated using the NLLS approach from the same data set. To simulate perturbed synthesis rates $\sigma_i'$, degradation rates $\delta_i'$ or initial abundances $a_{0i}'$ (to start from non-steady state conditions), we sampled gaussian noise such that ~5% of all genes are expected to be perturbed at most 2-fold. To simulate non-constant rates, we utilized the saturating function $f(t) = s \cdot c^{1-e^{-td}}$ to compute the synthesis or degradation rate at time $t$. To obtain new and old RNA values for each gene after 2 h of labeling, we numerically solved the ordinary differential equation for time-varying rates (see Supplementary Note 1). Here, $s$ is the unperturbed (start) state, $c$ is again gaussian noise as in the case of constant rate simulation, and we set $d = 1.39$. This function approaches $s \cdot c$ for $t \to \infty$, and our particular choice for $d$ let this function approach saturation with a speed such that half the way is achieved after 30 min. This procedure is implemented in the function *SimulateTimeCourseNonConstant*.

## Overview of kinetic modeling

The commonly used kinetic model of RNA expression assumes zero-order kinetics for RNA synthesis and first-order kinetics for degradation:

$$\frac{da}{dt} = \sigma - \delta a(t) \tag{2}$$

Here, $a(t)$ is the abundance of RNA at time $t$, and $\sigma$ and $\delta$ are the rate constants for synthesis and degradation, respectively. A gene is expressed at steady state if $\frac{da}{dt} = 0$, i.e. if $a(t) = \frac{\sigma}{\delta}$. The differential Eq. (2) can be solved for $a(0) = a_0$:

$$a(t) = \left(a_0 - \frac{\sigma}{\delta}\right)e^{-\delta t} + \frac{\sigma}{\delta} \tag{3}$$

Different variants of this model have been used to estimate RNA stability represented by the degradation rate $\delta$ or, equivalently, the RNA half-life $t_{1/2} = \log(2)/\delta$: (i) Finkel et al.[22] focused on unlabeled RNA and performed simple linear regression on Eq. (3) after log transformation and setting $\sigma = 0$. (ii) In Narain et al.[3], we used non-linear least

squares regression (NLLS) to fit the full model in Eq. (3), which has been done in a similar manner in Zuckermann et al.[18]. (iii) Boileau et al.[21] proposed to fit the full model using their pulseR package[19] based on the raw counts of reads showing T > C mismatches, and to remove bias of this approach using an additional nuisance parameter. (iv) Finally, we have presented a Bayesian method to estimate the degradation rate $\delta$ under steady state[17].

The main difference among methods (i)-(iv) is the error model employed. While (i) and (ii) assume homoscedastic gaussian errors of estimated pre-existing and newly synthesized RNA levels, either in log space (i) or of the levels directly (ii), pulseR models read counts using a negative binomial distribution assuming a gene specific overdispersion parameter that is jointly estimated from all samples. The Bayesian approach (iv) assumes that all data points are generated from a single degradation rate constant $\delta$ with the only source of error being the random sampling of T > C conversions.

## Kinetic model

To model the abundance of RNA at time $t$, $a(t)$, we define the following two functions for the abundances of old and new RNA, respectively, after labeling for time $t$ based on Eq. (3):

$$a^{(old)}(t; a_0, \delta) = a_0 e^{-t\delta} \tag{4}$$

$$a^{(new)}(t; \sigma, \delta) = \frac{\sigma}{\delta}\left(1 - e^{-t\delta}\right) \tag{5}$$

Under steady state assumptions, we have $a_0 = \frac{\sigma}{\delta}$, and can use the steady state function instead of Eq. (4):

$$a^{(old)}(t; \sigma, \delta) = \frac{\sigma}{\delta} e^{-t\delta} \tag{6}$$

We have data given in the form of total expression values $C_k$ and "measured" NTRs $\Xi_k$ for samples taken at time points $t_k$. Note that even if we use the same notation $C$ as for read counts above in "Read simulation", here we assume that $C_k$ is a normalized expression measure. We further index by $k$ to indicate the biological samples where data were obtained and drop the gene index $i$ for clarity. The NTRs $\Xi_k$ actually are not measured but are estimates of the parameter $ntr_k = \frac{a^{(new)}(t_k)}{a^{(new)}(t_k) + a^{(old)}(t_k)}$. We also have the beta approximation of the posterior distribution of $ntr_k$ given by $\alpha_k$ and $\beta_k$, i.e. $ntr_k | D \sim Beta(\alpha_k, \beta_k)$ for data $D$. We use bold face $\boldsymbol{t} = (t_1, \ldots, t_m)$, $\boldsymbol{\alpha} = (\alpha_1, \ldots, \alpha_m)$, $\boldsymbol{\beta} = (\beta_1, \ldots, \beta_m)$, etc. to denote the vector valued parameters.

## Modeling progressive labeling time courses

We define the random variables for old and new RNA as $O_k = C_k \cdot (1 - \Xi_k)$ and $N_k = C_k \cdot \Xi_k$. The distributions of $N_k$ and $O_k$ dependent on measurement noise from the sequencing experiment, uncertainty in the estimate $\Xi_k$ and biological variability. In a "progressive" labeling experiment each sample is labeled for a duration of $t_k$ starting from a common time point $t = 0$, and $\sigma$ and $\delta$ from Eqs. (2) and (3) remain constant after time $t = 0$. Then the expected values of $O_k$ and $N_k$ are

$$\mathbb{E}(O_k) = a^{(old)}(t_k; a_0, \delta) \tag{7}$$

$$\mathbb{E}(N_k) = a^{(new)}(t_k; \sigma, \delta) \tag{8}$$

All methods described below, except for pulseR, can conveniently be used in grandR using the function *FitKinetics*, which will fit either the

LM, the NLLS or NTR model for each gene by calling the respective functions mentioned below.

## LM method

For the linear model approach (method (i) in the text), we note that we have a linear function $\log(a^{(old)}(t;a_0,\delta)) = \log(a_0) - \delta t$ after log transforming Eqs. (4) or (6). Thus, $\delta$ and $a_0$ can be estimated using simple linear regression. Under the assumption of steady state, we can also obtain an estimate of $\sigma = a_0 \cdot \delta$. Note, however, that this assumes all $O_k$ to follow homoscedastic LogNormal distributions. We deem this model quite unrealistic, as at late time points $t_l \gg t_{1/2}$ (where $t_{1/2} = \log(2)/\delta$ is the half-life), $a^{(old)}(t_l; a_0, \delta)$ quickly approaches 0, and we therefore expect the residual $\log(a^{(old)}(t_l; a_0, \delta)) - \log(O_l)$ to be far greater than $\log(a^{(old)}(t_e; a_0, \delta)) - \log(O_e)$ at an earlier time point $t_e < t_{1/2}$. This approach is implemented by the *FitKineticsGeneLogSpaceLinear* function in grandR using the *lm* function of R. Confidence intervals are estimated using the *confint* function.

## NLLS method

For the non-linear least squares approach (method (ii) in the text), we assume $O_k$ and $N_k$ to be homoscedastic gaussian. Thus, $\sigma$, $\delta$ and $a_0$ (or $\sigma$ and $\delta$ under steady state assumptions) can be estimated using non-linear least squares regression. This is implemented in grandR by the function *FitKineticsGeneLeastSquares* using the *nls.lm* function from the minpack.lm package. Confidence intervals are estimated using *confint.nls.lm*.

## pulseR method

pulseR (method (iii) in the text) originally was developed for 4sU labeling experiments where labeled and unlabeled RNA was physically purified and sequenced separately[19]. It was later adapted to also handle nucleotide conversion RNA-seq data[21]. pulseR operates on labeled and unlabeled read counts (i.e. reads with and without observed T-to-C conversions), and includes additional nuisance parameters to model reads from unlabeled RNA with T-to-C conversions (e.g., sequencing errors) and reads from labeled RNA without T-to-C conversions (reads not covering 4sU incorporation sites). In our notation, the pulseR model is

$$a^{(unlabeled)}(t;a_0,\delta) = \mu_1 + a_0 e^{-t\delta} \tag{9}$$

$$a^{(labeled)}(t;\sigma,\delta) = \mu_2 + \frac{\sigma}{\delta}\left(1 - e^{-t\delta}\right) \tag{10}$$

Here, $\mu_1$ is the fraction of reads without T-to-C conversion, that indeed is not derived from old RNA, and $\mu_2$ is the fraction of reads with T-to-C conversions, that indeed is not derived from new RNA. Parameters are estimated using the counts of reads with and without T-to-C conversions instead of estimated old and new RNA levels $O_k$ and $N_k$ assuming reads to follow a negative binomial distribution with common dispersion parameter for a gene. This is implemented in grandR in the function *FitKineticsPulseR* using the code from ref. 21 provided on github (https://github.com/dieterich-lab/ComparisonOfMetabolicLabeling).

## NTR method

For the Bayesian NTR method (method (iv) in the text), we note that under the assumption of steady state $\delta = -\frac{1}{t}\log(1-ntr)$. Thus, the posterior distribution of the NTR given data, $ntr_k|D$, can be transformed into a distribution on $\delta$[17]. We assume $ntr_k|D \sim Beta(\alpha_k, \beta_k)$, and therefore the posterior density of the degradation rate is:

$$d(\delta;t_k,\alpha_k,\beta_k) = \frac{t}{B(\alpha_k,\beta_k)}\left(1 - e^{-t\delta}\right)^{\alpha_k-1} e^{-t\beta_k\delta} \tag{11}$$

By logarithmizing and setting the derivative to 0, we see that the MAP estimator is

$$\hat{\delta} = -\frac{1}{t}\log\frac{\beta_k}{\alpha_k + \beta_k - 1} \tag{12}$$

We can also transform the MAP estimator of $ntr_k|D$, yielding

$$\widetilde{\delta} = -\frac{1}{t}\log\frac{\beta_k - 1}{\alpha_k + \beta_k - 2} \tag{13}$$

Thus, transforming from the *ntr* parameter to the degradation rate $\delta$ results in non-invariance of the MAP estimator. Both estimators are implemented in grandR, and we chose to use the transformed NTR MAP estimator $\widetilde{\delta}$ by default. With several samples, the degradation rate is estimated by numerically maximizing the log posterior

$$g(\delta) = \sum_k (\alpha_k - 1)\log\left(1 - e^{-t_k\delta}\right) - t(\beta_k - 1)\delta \tag{14}$$

We use the *optimize* function built into R. For approximate $x\%$ credible intervals (CIs), we compute the critical drop in the log posterior distribution as $c = \frac{1}{2}\chi^2_{1,x}$ from a $\chi^2$ distribution with 1 degree of freedom similar to ref. 21. The rationale here is, that as we use a uniform prior, the posterior distribution is equal to the likelihood function. The CI is found by finding the values of $\delta$ left and right of the MAP estimate $\widetilde{\delta}$, where $g(\widetilde{\delta}) - g(\delta) = c$. For exact CIs, we numerically integrate $g$ using R's *integrate* function and report the $x\%$ CI interval with the MAP as the central point. This is implemented in grandR's function *FitKineticsGeneNtr*. It also provides an estimate of $\sigma = C_k\delta$.

## Temporal recalibration

grandR implements two ways to recalibrate labeling times. The first can only be used with progressive labeling data and makes use of the fact that our kinetic model poses some constraints on how the temporal dynamics can behave. For that, we fit the NLLS model simultaneously for all genes, and consider the labeling time as additional variables that are jointly optimized. To make this procedure more efficient and less prone to noise, we first make a rough estimate of the half-lives using the uncalibrated labeling times and use the top 200 expressed genes from the following half-life classes: 0–2 h, 2–4 h, 6–8 h, >8 h. Stratifying by half-life classes is important as many of the most strongly expressed genes have very long RNA half-lives. Importantly, however, the $n$ labeling time parameters can only be estimated up to a constant factor which corresponds to the time unit of the model. We make this model identifiable by assuming that the effective labeling time is equal to the nominal labeling time for the last time point. This procedure is implemented in the function *CalibrateEffectiveLabelingTimeKineticFit* in grandR.

The second method for temporal recalibration requires reference half-lives. For each biological sample the observed data can be transformed into half-lives for any labeling time (see below, "Transforming snapshot data"). We choose the labeling time such that the median log fold change between the reference and transformed half-lives across all genes is 0 by using the *uniroot* function of R. This procedure is implemented in the function *CalibrateEffectiveLabelingTimeMatchHalflives* in grandR.

## Transforming snapshot data

By solving Eqs. (4) and (5), we obtain

$$\delta = -\frac{1}{t}\log\frac{a^{(old)}(t)}{a_0} \tag{15}$$

$$= -\frac{1}{t}\log F \tag{16}$$

$$\sigma = a^{(new)}(t)\cdot\delta\cdot\frac{1}{1-e^{-t\delta}} \tag{17}$$

$$= -\frac{1}{t}a^{(new)}(t)\frac{\log F}{1-F} \tag{18}$$

with $F := \frac{a^{(old)}(t)}{a_0}$. To compute $\sigma$ and $\delta$ from this, the initial abundance $a_0$ at time $t=0$, i.e., at the start of labeling, must be known in addition to old and new RNA levels. This might not be the case, and only an abundance $a'$ at time $t'<0$ might be known, either by design of the experiments or because the effective labeling time is shorter than the nominal labeling time. In this case, the initial abundance can be computed as

$$a_0 = a'e^{t'\delta} + \frac{\sigma}{\delta}\left(1-e^{t'\delta}\right) \tag{19}$$

We use Eq. (5) to get rid of $\sigma$:

$$a_0 = a'e^{t'\delta} + a^{(new)}(t)\cdot\frac{1-e^{t'\delta}}{1-e^{-t\delta}} \tag{20}$$

Substituting this into Eq. (4):

$$a^{(old)}(t) = a'e^{(t'-t)\delta} + a^{(new)}(t)\cdot\frac{e^{-t\delta}-e^{(t'-t)\delta}}{1-e^{-t\delta}} \tag{21}$$

We solve this numerically for $\delta$ by using the R's *uniroot* function. Of note, this assumes $\sigma$ and $\delta$ to be constant throughout the time $[t',t]$. Transforming snapshot data is implemented in grandR's *TransformSnapshot* function.

## Old RNA fold changes

With a common reference sample with abundance $a_0$ and a common labeling time $t$, we can use Eq. (4) to derive the function that describes the change in degradation rates $\delta$ and $\delta'$ or RNA half-lives $t_{1/2}$ and $t'_{1/2}$ for an observed fold change of old RNA $f$ between two conditions with old RNA levels $a$ and $a'$:

$$a' = f\cdot a \tag{22}$$

$$\Longleftrightarrow e^{-t\delta'} = f\cdot e^{-t\delta} = e^{-t\delta + \log f} \tag{23}$$

$$\Longleftrightarrow \delta' = \delta - \frac{\log f}{t} \tag{24}$$

$$\Longleftrightarrow \frac{\log 2}{t'_{1/2}} = \frac{\log 2}{t_{1/2}} - \frac{\log f}{t} \tag{25}$$

$$\Longleftrightarrow t'_{1/2} = \frac{t_{1/2}}{1 - t_{1/2}\frac{\log f}{t}} \tag{26}$$

## Hierarchical Bayesian modeling of snapshot data

We define snapshot data for a single biological sample $k$ and a single gene from a nucleotide conversion RNA-seq experiment to be a tuple $D_k = (c'_k, c_k, u_{k,1},\ldots,u_{k,c}, tc_{k,1},\ldots,tc_{k,c})$. Here, $c'_k$ is the read count at the start of labeling at time $t=0$, and $c_k$ the read count at time $t$. For now, we ignore the need for normalization and assume that $c_k$ and $c'_k$ are directly comparable measures of gene expression, i.e. are already normalized. For ease of notation, we here assume a measurement $c'$ at $t=0$, but we can adapt our model in principle also to situations, where the measurement is taken at any time $t'$ (see above, *Transforming snapshot data*). $u_{k,r}$ and $tc_{k,r}$ for $r\in\{1,\ldots,c\}$ represent the number of uridines and the number of T-to-C conversions, respectively, for a read $r$, i.e. the sufficient statistics for estimation of the *ntr* parameter. We will omit the index $k$ if it is not necessary.

We assume that snapshot data $D$ are generated by the following process:

1. Sample the unobserved parameters $a_0$, $\sigma$ and $\delta$ from unknown distributions representing the biological variability of the true initial abundance, the synthesis rate and degradation rate, respectively; this uniquely determines the full temporal kinetics of the true RNA abundance $a(t)$ as well as $a^{(new)}(t)$ and $a^{(old)}(t)$ and $ntr(t) = \frac{a^{(new)}(t)}{a(t)}$.
2. Sample $c$ and $c_0$ from unknown distributions with mean $a(t)$ and $a(0)$, respectively. These distributions represent the technical noise of the measurement.
3. Sample $u_1,\ldots,u_c$ from the sequence of the gene. Which sequence is used depends on the protocol used for library preparation.
4. Sample $tc_r$ for $r\in\{1,\ldots,c\}$ from a binomial mixture distribution $BinomMix(u_r,p_e,p_c,ntr(t))$

Here, we are mainly concerned with snapshot data $\boldsymbol{D}^{A,B} = D_1^A,\ldots,D_n^A, D_1^B,\ldots,D_m^B$ involving several biological replicates from two conditions $A$ and $B$, and would like to infer the joint posterior distributions $\left(log_2\frac{\sigma_A}{\sigma_B}, log_2\frac{\delta_B}{\delta_A}\right)|\boldsymbol{D}^{A,B} = \left(log_2\frac{\sigma_A}{\sigma_B}, log_2\frac{HL_A}{HL_B}\right)|\boldsymbol{D}^{A,B}$. Note that for the synthesis rates we consider the log fold change $A$ vs $B$, i.e. $B$ is the *control* condition. We prefer to invert the log fold change of the degradation rates, which then corresponds to the more intuitive log fold change of the RNA half-lives $HL_A$ vs $HL_B$. Unfortunately, this is analytically intractable, and we found Markov chain Monte Carlo methods to be too inefficient considering the sheer size of $D$.

However, we show here that we can efficiently draw $N$ samples $(\sigma_1,\delta_1),\ldots,(\sigma_N,\delta_N)$ from the joint posterior $\sigma,\delta|\boldsymbol{D}$ for a single condition, with $n$ replicate samples, i.e. $\boldsymbol{D}=D_1,\ldots,D_n$. Hence, $\left(log_2\frac{\sigma_{A,1}}{\sigma_{B,1}}, log_2\frac{\delta_{B,1}}{\delta_{A,1}}\right),\ldots,\left(log_2\frac{\sigma_{A,N}}{\sigma_{B,N}}, log_2\frac{\delta_{B,N}}{\delta_{A,N}}\right)$ is a sample form the joint log fold change posterior distribution $\left(log_2\frac{\sigma_A}{\sigma_B}, log_2\frac{\delta_B}{\delta_A}\right)|\boldsymbol{D}^{A,B}$. To draw a single sample $\left(\sigma_j,\delta_j\right)$ from the posterior $\sigma,\delta|\boldsymbol{D}$, we consider the following processes separately:

1. Draw a sample $a'_j$ from the posterior distribution $a_0|\boldsymbol{D} = a(0)|\boldsymbol{D}$.
2. Draw a sample $a_j$ from the posterior distribution $a(t)|\boldsymbol{D}$.
3. Draw a sample $ntr_j$ from the posterior distribution $ntr(t)|\boldsymbol{D}$.

We then transform these samples into $\sigma$ and $\delta$ as described above under "Transforming snapshot data". Note that the prior distribution for $\sigma,\delta$ as well as $\left(log_2\frac{\sigma_A}{\sigma_B}, log_2\frac{\delta_B}{\delta_A}\right)$ is thereby implicitly defined by the priors for $a_0, a(t), ntr(t)$.

## Sampling from $\boldsymbol{a}(.)|\boldsymbol{D}$

We assume that read counts $c\sim NegBinom(\mu,d)$ are distributed according to a negative binomial distribution with mean $\mu$ and dispersion $d$. The dispersion parameter is defined as above such that the variance is $\mu+d\mu^2$. To enable efficient sampling, we assume $d$ to be fixed (for a single gene) and use *estimateDispersions* from the DESeq2

package for estimation. There is no obvious conjugate prior for $\mu$, however, we can reparametrize the negative binomial $NegBinom'(s,p)$ by $s = \frac{1}{d}$ and $p = \frac{s}{s+\mu}$. Then, $\mu = \frac{1-p}{p \cdot d}$.

It is easy to see that the Beta distribution is a conjugate prior for $p$: Given $n$ samples $\boldsymbol{c} = c_1, \ldots, c_n$, the density of the posterior for $p$ for a $NegBinom'(s,p)$ likelihood and $Beta(\alpha, \beta)$ prior is

$$\pi(p|\boldsymbol{c}) \propto \prod_k \left( \frac{\Gamma(s+c_k)}{\Gamma(s)\Gamma(c_k+1)} p^{c_k}(1-p)^s \right) \cdot \frac{\Gamma(\alpha+\beta)}{\Gamma(\alpha)\Gamma(\beta)} p^{\alpha-1}(1-p)^{\beta-1} \quad (27)$$

$$p^{(\alpha + \sum c_k)-1}(1-p)^{(\beta+ns)-1} \quad (28)$$

Thus, for the prior $p \sim Beta(\alpha, \beta)$, we have the posterior $p|c_1, \ldots, c_n \sim Beta(\alpha + \sum c_k, \beta + ns)$.

We use the full distribution of all genes to inform the prior distribution as follows. We first transform the expression value $c_i$ for each gene $i$ to $p_i = \frac{s_i}{s_i + c_i}$ with $s_i = \frac{1}{d_i}$ and use the method of moments to fit the hyperparameters $\alpha$ and $\beta$, which we then use for the whole data set of all genes.

So far, we have ignored normalization. For practical applications, this must be taken into account. We do this by the same approach as DESeq2, i.e. by rescaling read counts using a size factor to obtain normalized read counts[23]. This can be achieved in grandR by first calling the *Normalize* function, which places the normalized read counts into the default *data slot* of the grandR object.

Thus, to sample from $a(t)|D_1, \ldots, D_n$, we draw random numbers from a $Beta(\alpha + \sum c_k, \beta + \frac{n}{d})$ and to sample from $a_0|D_1, \ldots, D_n$, we draw random numbers from a $Beta(\alpha + \sum c_k', \beta + \frac{n}{d})$ distribution. Here, $c_k$ and $c_k'$ are the normalized read count from time $t$ and 0, respectively, of data set $D_k$, $d$ is the dispersion parameter estimated by DESeq2, and $\alpha$ and $\beta$ are the prior hyperparameters. Each of these Beta distributed values $p$ is then transformed via $\frac{1-p}{p \cdot d}$ to obtain a sample from $a(t)|D_1, \ldots, D_n$ or $a_0|D_1, \ldots, D_n$.

### Sampling from $\boldsymbol{ntr}(t)|\boldsymbol{D}$

The number of conversions on a read $tc_r \sim BinomMix(u_r, p_e, p_c, ntr)$ are distributed according to a binomial mixture distribution as defined above. The number of uridines $u_r$ is fixed, and to enable efficient sampling, we also assume the parameters $p_e$ and $p_c$ to be fixed. The posterior distribution $ntr|tc_1, \ldots, tc_r$ for a single biological sample, which is computed numerically by GRAND-SLAM, can be approximated by a Beta distribution, and we assume this Beta to be conjugate with the Beta prior used by GRAND-SLAM to compute the posterior distribution[17]. This posterior only quantifies technical variance of measuring the true $ntr$ for a single biological sample. To handle biological variability in addition, we introduce an additional hierarchical layer in our Bayesian model:

For each biological sample $k \in \{1, \ldots, n\}$, we have $ntr_k|\boldsymbol{D_k} \sim Beta(\alpha + \alpha_k, \beta + \beta_k)$. Here, $\alpha$ and $\beta$ are the parameters of the prior Beta distribution reflecting biological variability of $ntr$ across biological replicate samples and $\alpha_k$ and $\beta_k$ are the parameters estimated by GRAND-SLAM from the given $tc_{k,1}, \ldots, tc_{k,r}$, which reflect technical noise. The joint density of all $\boldsymbol{ntr} = (ntr_1, \ldots, ntr_n)$ is

$$\pi(\boldsymbol{ntr}|\alpha, \beta, \boldsymbol{D}) = \prod_k \mathcal{B}(\alpha + \alpha_k, \beta + \beta_k)^{-1} ntr_k^{\alpha + \alpha_k - 1}(1 - ntr_k)^{\beta + \beta_k - 1} \quad (29)$$

Here, $\mathcal{B}$ is the beta function. When imposing a prior on $(\alpha, \beta)$, the joint posterior of all parameters is

$$\pi(\boldsymbol{ntr}, \alpha, \beta|\boldsymbol{D}) \propto \pi(\alpha, \beta) \cdot f(\boldsymbol{ntr}|\alpha, \beta) \cdot f(\boldsymbol{D}|\boldsymbol{ntr}) \quad (30)$$

$$\propto \pi(\alpha, \beta) \cdot \prod_k \mathcal{B}(\alpha, \beta)^{-1} ntr_k^{\alpha-1}(1 - ntr_k)^{\beta-1} \quad (31)$$

$$\cdot \prod_k \prod_r (1 - ntr_k) B(tc_{k,r}; u_{k,r}, p_e) + ntr_k B(tc_{k,r}; u_{k,r}, p_c) \quad (32)$$

$$\tilde{\propto} \pi(\alpha, \beta) \cdot \prod_k \mathcal{B}(\alpha, \beta)^{-1} ntr_k^{\alpha-1}(1 - ntr_k)^{\beta-1} \cdot \prod_k (1 - ntr_k)^{\beta_k} ntr_j^{\alpha_k} \quad (33)$$

Here, $B(k; n, p) = \binom{n}{k} p^k (1-p)^{n-k}$, and the last line follows from our Beta approximation of the mixture model. Thus, the marginal posterior distribution of $(\alpha, \beta)$ is

$$\pi(\alpha, \beta|\boldsymbol{D}) = \frac{\pi(\boldsymbol{ntr}, \alpha, \beta|\boldsymbol{D})}{\pi(\boldsymbol{ntr}|\alpha, \beta, \boldsymbol{D})} \propto \pi(\alpha, \beta) \cdot \prod_k \frac{\mathcal{B}(\alpha + \alpha_k, \beta + \beta_k)}{\mathcal{B}(\alpha, \beta)} \quad (34)$$

If the marginal posteriors $ntr_k|\boldsymbol{D_k}$ overlap significantly, a point $ntr$ and, therefore, a $Beta(\alpha, \beta)$ prior with infinitesimally small variance or, equivalently, infinite $\alpha + \beta$ becomes probable. An appropriate constraint can be imposed using the prior distribution $\pi(\alpha, \beta)$. We decided to use the following sigmoid function

$$f_{o,s}(x) = \frac{1}{1 + e^{\frac{x-o}{s}}} \quad (35)$$

This can be integrated:

$$C_{o,s} = \int_0^\infty f_{o,s}(x) dx = s \cdot \log\left(1 + e^{\frac{o}{s}}\right) \quad (36)$$

and thus,

$$\pi(\alpha, \beta) = f_{o,s}(\alpha, \beta) \cdot C_{o,s}^{-1} \quad (37)$$

is a proper prior. $f_{o,s}$ is almost constant before the offset $o$ and quickly (depending on $s$) goes to zero after $o$, i.e. $o$ represents a maximal $\alpha + \beta$, or, equivalently, minimal variance, that has substantial prior probability. We set $o$ such that the variance of the prior $\pi(\alpha, \beta)$ is equal to the sample variance of $\frac{\alpha_1}{\alpha_1 + \beta_1}, \ldots, \frac{\alpha_n}{\alpha_n + \beta_n}$. Importantly, as long as (i) the mean $\frac{\alpha}{\alpha + \beta}$ is unconstrained and (ii) the minimal variance is constrained, the exact choice of the prior $\pi(\alpha, \beta)$ only has minor effect on sampling of $ntr|\boldsymbol{D}$.

To sample $ntr|\boldsymbol{D}$, i.e. the mean $\mu = \frac{\alpha}{\alpha + \beta}$ from the distribution $\pi(\alpha, \beta|\boldsymbol{D})$, we compute the marginal posterior on a grid of values[35]. Since we want to sample $\mu$, it makes sense not to build an $(\alpha, \beta)$-grid, but to reparametrize and build the grid with coordinates $(\log \frac{\alpha}{\beta}, \log(\alpha + \beta))$[35]. Note that $\log \frac{\alpha}{\beta} = logit(\mu)$. For each grid point $(x, y)$, we transform $\alpha = \frac{e^{x+y}}{e^x + 1}$ and $\beta = \frac{e^y}{e^x + 1}$, for which we compute the unnormalized posterior density defined in Eq. (34) with prior from Eq. (37), and, due to our reparametrization, multiply this by the Jacobian determinant

$$|J| = \left| \begin{pmatrix} \frac{e^{x+y}}{(e^x+1)^2} & \frac{e^{x+y}}{e^x+1} \\ -\frac{e^{x+y}}{(e^x+1)^2} & \frac{e^y}{e^x+1} \end{pmatrix} \right| \quad (38)$$

$$= \frac{e^{x+2y}}{(e^x+1)^3} + \frac{e^{2x+2y}}{(e^x+1)^3} \quad (39)$$

$$= \frac{e^{x+2y}}{(e^x-1)^2} \quad (40)$$

For numerical stability, we compute everything in log space, then subtract the maximal grid value and exponentiate[35]. To determine the grid bounds, we first find the maximum using R's *optim* function, and then go into positive and negative $x$ and $y$ directions to see where the grid would drop below 1000-fold of the maximal value using R' *uniroot* function. To sample $\mu$, we first sum over the columns of the grid and normalize to obtain a discrete probability distribution $l_1, \ldots, l_m$. Note that each $l_j$ corresponds to a particular value of $\log \frac{\alpha}{\beta}$. One of these values is sampled from the distribution $l_1, \ldots, l_m$ and random uniform jitter is added to fill the spacing of the grid[35]. This value $x$ is then transformed to $\mu = logit^{-1}(x)$.

### Reporting summary
Further information on research design is available in the Nature Portfolio Reporting Summary linked to this article.

### Data availability
All data sets used in this study are available in the in NCBI's Gene Expression Omnibus under GEO series accession codes GSE99970 (mESC pulse-chase data), GSE99972 and GSE99974 (mESC pulse data), GSE139151 (*NXF1* knockdown data), GSE162323 (SARS-CoV-2 data), and GSE155604 (*BANP* depletion data). All processed data (GRAND-SLAM outputs) are available on zenodo under https://doi.org/10.5281/zenodo. 7612564 (mESC pulse-chase data), https://doi.org/10.5281/zenodo.7630886 (mESC pulse data), https://doi.org/10.5281/zenodo. 5907183 (*NXF1* knockdown data), https://doi.org/10.5281/zenodo. 5834034 (SARS-CoV-2 data), and https://doi.org/10.5281/zenodo. 6976391 (*BANP* depletion data).

### Code availability
The grandR package (version 0.2.2) is available on github (https:// github.com/erhard-lab/grandR) and CRAN (https://CRAN.R-project. org/package=grandR) as open source (Apache License 2.0). R notebooks for generating all figures are provided in Supplementary Software 1. The notebooks and additionally data files containing precomputed simulated data sets are also available on zenodo (https:// doi.org/10.5281/zenodo.7843048).

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

## Acknowledgements

F.E. received funding by the Marie Skłodowska-Curie Actions Innovative Training Network VIROINF (grant agreement no. 955974), the Bavarian State Ministry of Science and Arts (Bavarian Research Network FOR-COVID), and the Deutsche Forschungsgemeinschaft (DFG, German Research Foundation) by project grant ER 927/2-1 and in the framework of the Research Unit FOR5200 DEEP-DV (443644894) project ER 927/4-1.

## Author contributions

T.R. and L.S. performed computational analyses and generated figures for the manuscript, F.E. performed analyses, generated figures, wrote the source code for the grandR package, and wrote the manuscript with contributions of all authors. All authors read and approved the final manuscript.

## Competing interests

The authors declare no competing interests.
