## [Peer Review File · Nature Communications]

grandR: a comprehensive package for nucleotide conversion
RNA-seq data analysisREVIEWER COMMENTS

Reviewer #1 (Remarks to the Author):

In this manuscript, the authors present grandR – a package for the analysis of metabolic labeling experiments, downstream of the GRAND-SLAM package that was created by the authors. This new package offers tools for quality control, calibration, differential gene expression analysis and inference of mRNA kinetics from specific types of metabolic labeling assays. The authors use simulated RNA-Seq data to develop their tool: evaluate several inference approaches, select number of replicates, timepoints and sequencing depth and develop a Bayesian inference model for transcription and degradation rates from single time-point labeling experiments. Using their tool, the authors analyze several public metabolic labeling datasets, and show that long 4sU labeling impacts cellular transcription, and the need for recalibration of early samples after 4sU introduction, which could be impacted by the rate of its transport into cells.

This work provides useful information for the design, quality control and analysis of metabolic labeling experiments which are becoming more and more popular. It also introduces improved modeling approaches and provides a streamlined tool for such analyses. While such tools have been presented before (e.g., pulseR, INSPECT), this work also provides additional advances.

However, there are also several concerns that need to be addressed.

1. Although the authors present a short overview of their tool in the first paragraph of the results, since the manuscript aims to present a new tool, it should include a more elaborate and detailed description of grandR: which inputs it requires? which experimental settings it is designed to analyze? What are its specific capabilities (quality controls available, visualizations, normalizations)? etc. For example, it is unclear which of the different kinetic modeling approaches (p. 3, “Comparative analysis of kinetic modeling”) is implemented in the tool?

In addition, a diagram that outlines all capabilities and available work-flows of grandR would be very helpful, and could replace, for example, figure 1A, which is currently showing a less informative screenshot of the tool.

2. A detailed comparison of the grandR tool to other computational tools available for analysis of metabolic labeling experiments (e.g., pulseR, INSPECT) is relevant. What are the similarities and differences and unique advantages of grandR? While the authors compare different kinetic models used by alternative tools, a more general comparison between the tools is relevant. For example, could grandR provide additional capabilities in analyzing mRNA kinetics beyond transcription and degradation, such as mRNA splicing, like other kinetic modeling tools offer (e.g., INSPECT)? Or, could grandR allow integration between multiple “snapshot” samples along an experimental timecourse, which if analyzed together, might improve model accuracy?

3. The kinetic inference offered by grandR tool is specifically designed to address two types of metabolic labeling experimental designs: progressive labeling and snapshot experiments. But the question of to what extent could this tool generalize to different designs of metabolic labeling experiments need to be addressed more thoroughly. In particular:

As the authors mention, other designs also exist such as pulse-chase labeling. While the authors discuss its disadvantages, could this design be analyzed using grandR?

Moreover, assumptions made in grandR could conflict with some experimental details and lead to biases. For example, grandR estimates transcription and degradation rates in snapshot experiments with a model that assume these rates are constant in time. This assumption is valid for short labeling

times, but if labeling time is prolonged and covers a biological response (non-steady state), these rates could change during that time, and model predictions might be affected.

4. The flow of the manuscript could be streamlined by separating sections that discuss analysis of data using the grandR tool, and reach conclusions on experimental design, and are therefore not directly related to grandR, and those sections that discuss simulation studies aimed at developing different parts of the algorithm.

5. "Choosing number of replicates, time points and sequencing depth". It would be interesting to compare between experimental designs with an overall identical number of samples, but that are either spread throughout the timecourse (higher resolution) or repeatedly sample fewer timepoints (replicates).

Reviewer #2 (Remarks to the Author):

Rummel et al. present grandR, a software package for nucleotide conversion sequencing data. This is an important contribution to the scientific literature, since methods like (single-cell) SLAM-seq will certainly gain in popularity in the future due to the ability to add temporal information to RNA-seq data. A software package like this can contribute to make nucleotide conversion sequencing data accessible to a wider range of users. For single-cell RNA-seq, packages like Seurat or Scanpy have greatly contributed to the spread of the technology, and grandR has the potential of achieving a similar goal for nucleotide conversion data.

While grandR is certainly a powerful and important package, it would be important to restructure the manuscript in order to make the underlying assumptions, concepts and models more accessible. Right now, the manuscript mostly shows applications of the package, including on simulated data. By contrast, it is explained only very briefly, and often in very technical terms, how the analysis works. While the level of mathematical detail given in the Methods is appropriate, it would be important to give more intuitive explanations for a wider audience in the main text.

We also tested the software package, and we were happy to see that the installation went very well and reasonably fast. We ran two of the provided vignettes, the sample data was available, downloaded fast and computed without issues. All of the plots generated looked similar to the examples given, based on the same data. Comments and questions regarding the software are listed below.

Major comments

1. Large parts of the applications shown focus on quality control or on technical aspects of SLAM-seq data (effects of long-term labeling, temporal recalibration, number of replicates, different kinetic models). While this is all useful, the manuscript does not really demonstrate yet which type of biological findings can be achieved with single-cell SLAM-seq. In particular for scientists that are not doing SLAM-seq experiments yet, it might be useful to demonstrate which types of analysis can be done with grandR by reanalyzing published single-cell SLAM-seq data (e.g. Erhard et al., 2019 or Cao et al., 2020). For these published datasets it would not be about generating new findings, but about demonstrating the use of grandR on real data. Related to this, the manuscript does not mention applicability to single-cell data – it would be important to describe how grandR can be used for single-cell SLAM-seq data.

2. As mentioned in my general summary above, the level of explanation of hypotheses, concepts and

models is often not well balanced. The general ideas are too "hidden" and not emphasized enough in the main text. This leads to an unclear message, and one really needs to dive into the methods section to understand the key concepts. Namely:

a. One example for this is Eq. 3, which is basically not explained (there might be more detail in the methods, but it's hard to find). The authors here aim to introduce a non-steady state model, which would be very useful for many applications, but as presented this is very difficult to understand. The authors should explain the equation, and importantly they should also discuss underlying assumptions and limitations. Does this hold for all non-steady state cases (the example in Fig. 1c is "almost" steady-state), including rapid decay or non-monotonic dynamics?

b. Another example concerns time calibration. This is an appealing idea, but as mentioned in the Methods section: 'grandR implements two ways to recalibrate labeling times. The first can only be used with progressive labeling data and makes use of the fact that our kinetic model poses some constraints on how the temporal dynamics can behave.' What are these constraints?

3. I found the improvement of the Bayesian hierarchical model compared to the naive approach rather unspectacular: For the synthesis rate there is basically no improvement, and for the degradation rate, as the authors discuss, according to Eq. 3 the correlation shown in Fig. 5D is not the right way to look at the data. The authors should discuss the advantages of their Bayesian model for analysis of snapshot experiments in more detail.

4. I was confused about the analysis regarding the required number of replicates. Since this was done on simulated data, it is not clear how this relates to real experiments that will be affected by biological sources of variation. This would warrant a discussion of which sources of variation between replicates are included here, and which ones are not. As presented, the usefulness for an experimental lab is unclear (and the suggested number of 6-12 samples per condition will often be unrealistically high). Also, can the authors give an intuition on the relationship between number of replicates and sequencing depth that they observe?

5. Software package:

General points:

a. The expected input does not seem to be very well defined (apart from: "e.g. processed by the GRAND-SLAM pipeline"). Since GRAND-SLAM seems to be the only input data that is confirmed to work, GRAND-SLAM is basically an integral part of the pipeline, which leads to the next point:

b. The GRAND-SLAM pipeline is somewhat hard to obtain, requiring a form with first and last name, institution and institutional email that looks to be manually checked by an employee who then sends out an email containing a download link. This is not the way to distribute a 'quasi-standard' software package (which the lab seems to be aiming for). Working examples available through repositories could be Seurat or Scanpy (and its offspins).

c. After obtaining the GRAND-SLAM software it came without any documentation – no licence, link to documentation or a 'getting started'/'readme'-type of document. This makes it a bit difficult to use. It's also provided as compiled java code, which is maybe not ideal for scientific software.

Specific points:

d. The dependencies need to be amended to include DESeq2 (which is used in some functions)

e. SetParallel(cores = n) did not work for us (Xubuntu 20.04.3; R 4.1.1, 12 virtual cores). Could be an issue when working on a server.

f. It would be helpful if the gene names were included (e.g. as a title) in plots that showcase dynamics of individual genes, currently there does not seem to be an option to do this.

g. Saying the workflow 'advocates' systematic sample names (first paragraph Results) is a bit of an understatement - the package does likely not work otherwise.

Minor comments

6. The synthesis and degradation rates are supposed constant, which is probably a good approximation. Can the model handle (slowly) varying rates?

7. Some abbreviations (most notably NTR) are not formally introduced, decreasing readability

8. Results in Fig. 3a are described as 'significant' in the text, but I can't find p-values to back up the claim

9. Ref. 14 is now published (Cao et al., Nature biotechnology, 2020)

10. Can the authors please add a reference about the active transport of 4sU across cell membranes?

11. In the third paragraph of the Results, the authors states that 4sU treatment for > 4h is problematic, but the 4h threshold is only proven in the following abstract. This is a bit confusing to read, and the recommendation not to go beyond 4h should come after discussing Fig. 2c.

Reviewer #3 (Remarks to the Author):

In the manuscript "GrandR: a comprehensive package for nucleotide conversions sequencing data analysis," Rummel et al. address the experimental and computational challenges of studying the temporal dynamics of gene expression by metabolic RNA labeling and sequencing upon chemical nucleotide conversion (i.e. SLAM-seq, Timelapse-seq or TUC-seq). To this end, the authors assembled a bioinformatic analysis workflow (grandR) that incorporates previously described computational methods (i.e., GRAND-SLAM) and well-established kinetic models to re-analyze previously published nucleotide conversion datasets.

While on the technical side, grandR may facilitate the analysis of nucleotide conversion sequencing data to measure cellular gene expression kinetics, the findings are limited in overall scope and do not provide substantial new technical or biological insights that would merit publication in Nature Communications. In fact, much of the parameters that are reported to impact the accuracy of mRNA half-live estimation strongly depend on the chosen experimental context (i.e., cell line, labeling conditions, library preparation protocol, global RNA metabolic state, etc.) and warrant careful experimental validation prior to drawing biological conclusions, an aspect that is rightly-so not addressed in this manuscript. The comments below should be considered by the authors to increase transparency and clarity of their workflow and benchmark the performance of grandR in light of substantially varying technical parameters using appropriately controlled experimental datasets.

Major comments:

For benchmarking of grandR, the authors build entirely on previously published datasets, that presumably vary in technical aspects, including library preparation protocol, cell type, and labeling conditions (i.e. labeling efficiency and timepoints). But many of these technical parameters remain poorly addressed. How does grandR perform on datasets acquired in the same cell type but varying

4sU concentrations? In the absence of such a reference dataset, much of the presented analysis is rather circumstantial and does not allow to draw major conclusions. As a consequence, the statement that "these analyses advocate against long-term 4sU treatment beyond 4h even with low doses of 4sU and, therefore, argue against pulse-chase designs for estimating RNA half-lives in general" is misleading.

The details on pre-processing steps, mapping parameters, chosen annotations etc. are largely missing in the methods section. In the absence of this information (and careful benchmarking) it is difficult to interpret the presented results. Furthermore, at least some of the analyzed data represent QUANTseq libraries (generated in at least two different cell types; mouse mESC, human MCF7) heavily rely on accurate annotation of mRNA 3' ends (which vary substantially between cell types), and preprocessing (i.e., trimming of random forward priming events, which may introduce errors in T>C conversion rate estimates). Do the presented data show per-gene or per-transcript 3' isoform analyses? If per-gene analyses, how was the signal averaged across different isoforms?

While a long-term effect of 4sU labeling on gene expression (shown in Fig.2) is not entirely unexpected (given the reduction in cell viability to 80%), the effect on half-life calculation remains somewhat unclear. How do genes called differentially expressed at the 24h labeling timepoint (based on a very loose cutoff) vary in apparent expression along the entire timecourse? How do principal component analyses shown in Fig. 2A look like when including all labeling timepoints? Similarly, how does this analysis look like for all other published datasets? And how does half-life estimation change when excluding stepwise later labeling timepoints (particularly when investigating long-lived transcripts, for which accurate prediction relies on late measurement timepoints)?

While the analysis in Fig. S1C argues that short half-lived mRNAs (<5 h) are accurately modeled even in the absence of late measurement timepoints, this seems to not hold true for long-lived transcripts. Hence, the recommendations provided in the manuscript to measure (progressively) at early metabolic labeling timepoints would only hold for short-lived but not for long-lived transcripts. In fact, this is also observed in their simulated data (Fig. 3D) and hence, should be put in context.

The introduced analysis methods are tested/evaluated with in-silico datasets that were created with a simple simulation approach. It is, however, unclear how well these simulations reflect real data: how do simulated distributions of T-to-C conversions compare to real datasets? How does their method compare to other, established RNA-seq simulators (except for the missing T-to-C conversion feature)? Is their method producing realistic half-life distributions?

The authors state in their introduction that "the percentage of such [i.e.4sU-labeled] reads is in the order of 20-80% and depends on the ratio of 4sU and normal uridine available for incorporation into nascent RNA, the read length, and the uridine content of RNA." But in the presented re-analysis of published datasets and in their data simulations, the impact of these parameters on the performance of their analysis workflow is largely ignored. How does read-length (and U-content) influence robustness of their presented analysis workflow? What is the limit of 4sU-incorporation needed in order to draw meaningful biological conclusions? Such parameters seem as important for robust data acquisition as number of replicates or progressive timepoints.

While the proposed temporal recalibration model obviously increases the model fit, it is less clear whether it results in more accurate half-life predictions. Did the authors compare corrected halflives to published/validated datasets or the mentioned 'reference RNA half-lives'? The authors should also consider testing whether their proposed temporal recalibration model improves correlation of halflives between replicated data. We furthermore somewhat disagree with the used formulation that for earlier timepoints the effective 'labeling time' is expected to be much shorter than the nominal time. Instead, we believe that the effective concentration of 4sU in the nucleus is increasing over time (until saturation) and thus the probability of a U being labeled increases over time. The authors should reformulate and test whether their assumption is true, e.g., by analyzing data from a 4sU chase

labelling experiment in which cells are typically exposed to 4sU until full saturation. Or alternatively by providing an experimental estimate for labeling density. In this case they should not see the same effects on the residuals as for the pulse time course shown in Figure 4. They might also consider building such a gradual increase in effective 4sU concentration into their simulation approach to make it more realistic.

Minor comments:

In the introduction the authors state that 4sU "is actively transported across cell membranes via nucleoside transporters." Can the authors provide a reference for this statement? Prior work in yeast suggests that 4sU mainly enters cells via passive diffusion mediated by nucleoside equilibrate transporters of the Ent family.

Introduction: "A limitation of 4sU conversion approaches is that concentrations of 4sU that are tolerated by cells commonly only replace 1 in 40 uridines by 4sU." Where does this estimation come from?

NTR is being used inconsistently and is not well defined in the manuscript.

REVIEWER COMMENTS

Reviewer #1 (Remarks to the Author):

In this manuscript, the authors present grandR – a package for the analysis of metabolic labeling experiments, downstream of the GRAND-SLAM package that was created by the authors. This new package offers tools for quality control, calibration, differential gene expression analysis and inference of mRNA kinetics from specific types of metabolic labeling assays. The authors use simulated RNA-Seq data to develop their tool: evaluate several inference approaches, select number of replicates, timepoints and sequencing depth and develop a Bayesian inference model for transcription and degradation rates from single time-point labeling experiments. Using their tool, the authors analyze several public metabolic labeling datasets, and show that long 4sU labeling impacts cellular transcription, and the need for recalibration of early samples after 4sU introduction, which could be impacted by the rate of its transport into cells.

This work provides useful information for the design, quality control and analysis of metabolic labeling experiments which are becoming more and more popular. It also introduces improved modeling approaches and provides a streamlined tool for such analyses. While such tools have been presented before (e.g., pulseR, INSPECT), this work also provides additional advances.

We thank the reviewer for this positive assessment.

However, there are also several concerns that need to be addressed.

1. Although the authors present a short overview of their tool in the first paragraph of the results, since the manuscript aims to present a new tool, it should include a more elaborate and detailed description of grandR: which inputs it requires? which experimental settings it is designed to analyze? What are its specific capabilities (quality controls available, visualizations, normalizations)? etc. For example, it is unclear which of the different kinetic modeling approaches (p. 3, “Comparative analysis of kinetic modeling”) is implemented in the tool?

All capabilities of grandR are described in the R package documentation and its vignettes, and we felt that our manuscript is not the right place to describe everything in detail. However, the reviewer is absolutely right that the overview of our package in the manuscript was too short and did not give a good impression about its capabilities. We now expanded the first subsection of the Results to provide a more comprehensive overview of grandR. We also believe that this will now make it easier for the reader to cross-reference what he or she reads in the paper and what is found in the package and its documentation.

In addition, a diagram that outlines all capabilities and available work-flows of grandR would be very helpful, and could replace, for example, figure 1A, which is currently showing a less informative screenshot of the tool.

We initially also thought about such a diagram as proposed by the reviewer. However, this diagram either was too complex, or it was oversimplistic and gave the wrong impression about how comprehensive grandR is. Fig. 1A goes into the more simplistic direction, with the additional benefit that by showing actual code, it is made clear that also non-experts in programming will be able to use our package for data analysis (which is an important message we want to convey). However, we agree with the reviewer that a more comprehensive overview of the capabilities of grandR beyond the new textual description in the manuscript would be helpful. We therefore designed a cheatsheet summarizing the most relevant capabilities and how they are practically used in grandR. Such cheatsheets have become quite popular for famous and widely used R packages such as ggplot2, and we believe that this is the right way to present the capabilities to users of our package.

2. A detailed comparison of the grandR tool to other computational tools available for analysis of metabolic labeling experiments (e.g., pulseR, INSPEct) is relevant. What are the similarities and differences and unique advantages of grandR? While the authors compare different kinetic models used by alternative tools, a more general comparison between the tools is relevant. For example, could grandR provide additional capabilities in analyzing mRNA kinetics beyond transcription and degradation, such as mRNA splicing, like other kinetic modeling tools offer (e.g., INSPEct)? Or, could grandR allow integration between multiple “snapshot” samples along an experimental timecourse, which if analyzed together, might improve model accuracy?

We added an additional paragraph to the introduction to highlight the main difference between grandR and all existing tools: grandR is developed for nucleotide conversion data, whereas all other tools expect data after affinity purification. We believe that a comparison of each package’s capabilities in form of a table as in the pulseR paper would give the wrong impression that all these packages are alternative ways to solve the same problem. Nevertheless, we agree with the reviewer that differences in the different packages should be emphasized. We thus rewrote large parts of the discussion. Specifically, we clearly discuss the only limitation of grandR we see compared to other packages (it does not implement methods to analyze RNA processing, since affinity purification data is more suitable for this), and we provide more discussion about the benefits of multiple snapshots.

3. The kinetic inference offered by grandR tool is specifically designed to address two types of metabolic labeling experimental designs: progressive labeling and snapshot experiments. But the question of to what extent could this tool generalize to different designs of metabolic labeling experiments need to be addressed more thoroughly. In particular:

As the authors mention, other designs also exist such as pulse-chase labeling. While the authors discuss its disadvantages, could this design be analyzed using grandR?

We thank the reviewer for this comment. Even though we never perform pulse-chase experiments ourselves, this does not mean that there are no situations where this might be advantageous. For this reason, we now implemented this in grandR. We added a vignette to show this, and we indicate this clearly in the text in the overview of grandR and when we make the conclusion to be careful with pulse-chase designs. We used this functionality now implemented in grandR to perform the comparative analyses of RNA half-life estimates (with and without recalibration for progressive labeling, pulse-chase, Actinomycin-D chase), as suggested by reviewer 3.

Moreover, assumptions made in grandR could conflict with some experimental details and lead to biases. For example, grandR estimates transcription and degradation rates in snapshot experiments with a model that assume these rates are constant in time. This assumption is valid for short labeling times, but if labeling time is prolonged and covers a biological response (non-steady state), these rates could change during that time, and model predictions might be affected.

We fully agree with the reviewer that non-constant rates and prolonged labeling are in conflict with the employed model. We also would like to point out that non-steady state situations, where e.g. synthesis rates changed due to activation of specific transcription factors (e.g. as a biological response to an external stimulus) can nevertheless be modeled with constant rates (e.g. one specific rate before, one specific rate after the stimulus). Gradually changing rates (that go beyond non-steady state situations with an abrupt change in the rate), would be approximated by a piecewise constant function in grandR, when there are multiple snapshots over time. It is important to note that also tools such as INSPEcT, which can be used with non-constant rates, are based on similar approximations (in case of INSPEcT, it is piecewise linear functions). In principle the models used in grandR can be adapted e.g. to use piecewise linear functions or any predefined parametric family of functions defined over the whole experimental time. We believe that our Bayesian hierarchical model provides a nice framework to explore such possibilities in the future. Here, we mainly developed this model for a different purpose, which is to estimate fold changes of synthesis and degradation rates upon a perturbation as in Muhar et al., Science (2018). We now added more discussion on this topic.

4. The flow of the manuscript could be streamlined by separating sections that discuss analysis of data using the grandR tool, and reach conclusions on experimental design, and are therefore not directly related to grandR, and those sections that discuss simulation studies aimed at developing different parts of the algorithm.

We tried different flows for our manuscript and decided that we would like to leave it as it is. We believe that having several parts, each describing a new concept/approach in grandR and coming to a clear conclusion, is much better than taking apart these parts. We also believe that the order of these parts makes sense:

1. We start with the toxicity analysis, which is also the first step of any analysis workflow. This part has a clear conclusion (“be careful and test for these effects in your data”).
2. After that we go into progressive labeling designs:
 - a. comparison based in simulated data using one specific setting of # replicates, time points, read counts etc. -> clear ranking of methods
 - b. then expand to many different settings -> it’s important how time points are chosen
 - c. finally, we apply it to real data, which we show requires recalibration of labeling times.
3. After that, we go into snapshots:
 - a. first comparison of fold changes of labeled or unlabeled RNA to fold changes of estimated rates
 - b. then comparing the Bayesian approach to P values from labeled or unlabeled RNA
 - c. finally, application of this approach to real data (which involved our second temporal recalibration method).

The main disadvantage we see with this structure is that the need for temporal recalibration is introduced as part of progressive labeling designs. However, introducing this earlier is not possible, since the reader must already be familiar with the model and progressive labeling to understand the analyses done in this section. Introducing it later has the disadvantage that the analysis of real progressive labeling experiments would be separated from the simulation study of progressive labeling.

To ensure a better flow, we greatly expanded the first section, providing a more comprehensive overview of the capabilities of grandR, and we took out a lot of unnecessary technical detail from the

main text (or moved it to the Methods section when it was appropriate), and provide more high-level description.

5. “Choosing number of replicates, time points and sequencing depth”. It would be interesting to compare between experimental designs with an overall identical number of samples, but that are either spread throughout the timecourse (higher resolution) or repeatedly sample fewer timepoints (replicates).

We agree that this would be an interesting analysis. However, there are many combinations of such configurations that would be interesting to compare. What we decided for here actually is very close to the suggestion of the reviewer:

- Figure 3D compares settings with either around 6 or around 12 samples (instead of an overall identical number of samples). We wanted to use previously utilized labeling times and therefore restricted this analysis to 0,1,2,4, and 8h labeling samples. It is “around” 6 or 12, since we wanted to maintain the same number of replicates per time point.
- Figures S3B-C compares situations with fewer replicates (or fewer reads per sample).

We realized that we did not include a description what “2x 0+1+2h” etc means. We now added a sentence to the figure legend to make this clear. It is important to note that any comparison desired can easily be performed by using grandR, which includes documented function for simulation of experiments, and we want to emphasize that R notebooks are provided to generate these figures, where it is straight forward to change the numbers.

Reviewer #2 (Remarks to the Author):

Rummel et al. present grandR, a software package for nucleotide conversion sequencing data. This is an important contribution to the scientific literature, since methods like (single-cell) SLAM-seq will certainly gain in popularity in the future due to the ability to add temporal information to RNA-seq data. A software package like this can contribute to make nucleotide conversion sequencing data accessible to a wider range of users. For single-cell RNA-seq, packages like Seurat or Scanpy have greatly contributed to the spread of the technology, and grandR has the potential of achieving a similar goal for nucleotide conversion data.

We thank the reviewer for this very positive assessment, and we are happy that the reviewer recognized our goal to provide a standard analysis package for nucleotide conversion data similar to what the fantastic Seurat and Scanpy packages are for single cell data.

While grandR is certainly a powerful and important package, it would be important to restructure the manuscript in order to make the underlying assumptions, concepts and models more accessible. Right now, the manuscript mostly shows applications of the package, including on simulated data. By contrast, it is explained only very briefly, and often in very technical terms, how the analysis works. While the level of mathematical detail given in the Methods is appropriate, it would be important to give more intuitive explanations for a wider audience in the main text.

We thank the reviewer for this comment. We made several changes to the Results section to remove technical details when appropriate, and to ensure a better flow. Specifically:

- We expanded the overview of grandR to give a better impression of its capabilities.

- We completely rewrote the first part of “Comparative analysis of kinetic modeling”, removing all technical details and providing explanation on the kinetic model, the four methods and our read simulation.
- We removed all technical details from “Estimating changes in synthesis or degradation from snapshot experiments”
- Based on several other comments by all reviewers, we provide more explanations throughout the results section

We believe that these changes greatly enhanced the readability of our manuscript.

We also tested the software package, and we were happy to see that the installation went very well and reasonably fast. We ran two of the provided vignettes, the sample data was available, downloaded fast and computed without issues. All of the plots generated looked similar to the examples given, based on the same data. Comments and questions regarding the software are listed below.

Major comments

1. Large parts of the applications shown focus on quality control or on technical aspects of SLAM-seq data (effects of long-term labeling, temporal recalibration, number of replicates, different kinetic models). While this is all useful, the manuscript does not really demonstrate yet which type of biological findings can be achieved with single-cell SLAM-seq. In particular for scientists that are not doing SLAM-seq experiments yet, it might be useful to demonstrate which types of analysis can be done with grandR by reanalyzing published single-cell SLAM-seq data (e.g. Erhard et al., 2019 or Cao et al., 2020). For these published datasets it would not be about generating new findings, but about demonstrating the use of grandR on real data. Related to this, the manuscript does not mention applicability to single-cell data – it would be important to describe how grandR can be used for single-cell SLAM-seq data.

We would like to make clear that this manuscript is not focused on single cell applications. In fact, we are in contact with many researchers who use our GRAND-SLAM software who are only interested to apply nucleotide conversion sequencing in bulk and not in single cells. There is no question that there are also many researchers interested in the single cell approach of it, but there is also a huge demand for standard tools to analyze bulk data. This is what we focus on in this manuscript. We feel that including specific methods for single cell approaches would be premature (they are still under active development by us and others), and would also not fit into the scope of this manuscript.

Nevertheless, we agree fully with the reviewer that this is a timely and important aspect. We therefore added additional functionality to grandR for processing single cell data as well. A typical workflow is to load single cell data into grandR, visualize QC metrics with regard to labeling, and then hand over data, e.g. in form of both total and new RNA as two data modalities to Seurat to perform single cell data analysis. In the manuscript, this is briefly described in the overview section, and we provide a vignette on this, processing the Cao et al data, as suggested by the reviewer.

2. As mentioned in my general summary above, the level of explanation of hypotheses, concepts and models is often not well balanced. The general ideas are too "hidden" and not emphasized enough in the main text. This leads to an unclear message, and one really needs to dive into the methods section to understand the key concepts. Namely:

a. One example for this is Eq. 3, which is basically not explained (there might be more detail in the methods, but it's hard to find). The authors here aim to introduce a non-steady state model, which would be very useful for many applications, but as presented this is very difficult to understand.

The authors should explain the equation, and importantly they should also discuss underlying assumptions and limitations. Does this hold for all non-steady state cases (the example in Fig. 1c is "almost" steady-state), including rapid decay or non-monotonic dynamics?

As indicated above, we have rewritten this part. Now the formula is not introduced anymore in the results section. We also added a detailed discussion on non-steady state cases and on the limitations of this model when rates are not constant.

b. Another example concerns time calibration. This is an appealing idea, but as mentioned in the Methods section: 'grandR implements two ways to recalibrate labeling times. The first can only be used with progressive labeling data and makes use of the fact that our kinetic model poses some constraints on how the temporal dynamics can behave.' What are these constraints?

We now explain the rationale behind this more clearly in the results section:

“The effective labeling time for a sample is a global parameter that is common to all genes. Moreover, the temporal behavior of new and old RNA for all genes is constrained by the kinetic model: The new RNA levels from different time points lie on a single curve (the model fit) that approaches a specific steady state level. Conceptually, if the effective labeling time for a 1h sample actually is 40 minutes, the new RNA level for this sample for each gene would be below the model fit, and moving this sample to the 40 minute time point would correct for that (Fig 4A). Thus, we reasoned that it should be possible to estimate effective labeling times by maximizing the joint likelihood of all gene specific synthesis and degradation rates and the effective labeling times.”

3. I found the improvement of the Bayesian hierarchical model compared to the naive approach rather unspectacular: For the synthesis rate there is basically no improvement, and for the degradation rate, as the authors discuss, according to Eq. 3 the correlation shown in Fig. 5D is not the right way to look at the data. The authors should discuss the advantages of their Bayesian model for analysis of snapshot experiments in more detail.

The reviewer is right that the estimated change of the synthesis rate is very similar to the observed change in new RNA in situations where indeed only synthesis rates are changed. However, we also show that a change in RNA stability can induce changes in new RNA (especially for genes with short RNA half-lives). We now make this more explicit by saying “Thus, if synthesis rates are changed, these changes are directly reflected on the change in new RNA. However, the converse is not true since a change in new RNA levels can also be due to a change in RNA stability especially for short-lived RNAs.” In addition, as suggested by the reviewer, we provide more discussion why estimating the uncertainties in the rate parameters is essential for snapshot experiments:

“Here we have shown that instead of progressive labeling designs, also a single time point (“snapshot”) can be used to infer the kinetics of RNA expression. The main challenge for that is to quantify the uncertainties associated with parameters estimated from snapshots: Under several circumstances, these estimates can be highly inaccurate: If the number of reads for a gene is low, or the overall 4sU incorporation rate in an experiment is weak, the estimated percentage of labeled RNA (NTR) cannot be estimated accurately¹⁵, which directly results in inaccurate estimates of the kinetic parameters. This problem is exacerbated if the labeling time does not match the gene specific RNA half-life. For instance, if a sample was labeled for 1h, the NTR is roughly 8.3% for an RNA half-life of 8h. If the estimated NTR is only slightly inaccurate, the estimated half-life will deviate substantially from 8h. Our Bayesian hierarchical model therefore does not only provide point estimates of the

kinetic rate parameters, but also estimates their full posterior distributions, which reflect these uncertainties.”

4. I was confused about the analysis regarding the required number of replicates. Since this was done on simulated data, it is not clear how this relates to real experiments that will be affected by biological sources of variation. This would warrant a discussion of which sources of variation between replicates are included here, and which ones are not. As presented, the usefulness for an experimental lab is unclear (and the suggested number of 6-12 samples per condition will often be unrealistically high). Also, can the authors give an intuition on the relationship between number of replicates and sequencing depth that they observe?

The main point of this part is that depending on the actual RNA half-life of a gene, different labeling times are important. We agree that real experiments are more complicated and that the accuracies will also depend on biological variability.

We actually simulated biological variation, which is now also clearly stated in the manuscript. We added more explanation and also clearly state the caveats of this analysis:

“Interestingly, increasing the number of replicates per time point boosted the accuracy stronger than increasing the number of reads. This likely is an effect of the biological variability that we simulated. With biological variability the actual RNA half-life is not the same but varies slightly for a gene among all samples (time points and replicates). Additional replicates are thus likely important to accurately include the variability into the kinetic model. Importantly, however, the extent of improvement due to more replicates depends on the magnitude of biological variation. In general, the reported accuracies of the estimates depend on the assumptions made for simulation and the parameters used. We therefore recommend running custom simulations using the functions implemented in grandR to come to an informed decision for planning experiments.”

5. Software package:

General points:

a. The expected input does not seem to be very well defined (apart from: "e.g. processed by the GRAND-SLAM pipeline"). Since GRAND-SLAM seems to be the only input data that is confirmed to work, GRAND-SLAM is basically an integral part of the pipeline, which leads to the next point:

This was not made clear in the text. grandR is agnostic of its input. The basic input required is a count matrix, and a matrix containing NTR values, and no matter where these matrices came from, the available methods can be applied to them. For convenience, grandR provides functions to read these data from defined input files. Not surprisingly, we favor GRAND-SLAM to produce the input for grandR, but it is not an integral part of the pipeline. In fact, in the vignette we now added for the analysis of single cell data, we demonstrate this: It loads the count matrices of total and new RNA provided by Cao et al. directly from the GEO repository.

We describe this more clearly in the overview section of the results.

b. The GRAND-SLAM pipeline is somewhat hard to obtain, requiring a form with first and last name, institution and institutional email that looks to be manually checked by an employee who then sends out an email containing a download link. This is not the way to distribute a 'quasi-standard' software package (which the lab seems to be aiming for). Working examples available through repositories could be Seurat or Scanpy (and its offspins).

This manuscript is not about our already published GRAND-SLAM tool. grandR is freely and easily available. Also, as pointed out above, GRAND-SLAM is not required to run grandR.

There is a reason why GRAND-SLAM is only available after agreeing to a license and leaving contact details: We have a patent pending on GRAND-SLAM. For academia, it is freely available, but we are legally bound by the University of Würzburg to distribute it in this way. We would also like to point out that this is not unique to GRAND-SLAM, e.g. the extremely popular MaxQuant software for massspec data analysis is also only available after leaving contact details and agreeing to its license.

However, for several reasons, the University of Würzburg and we decided not to further pursue this patent. Once the University gives up on the protection, GRAND-SLAM will be made available at github.

c. After obtaining the GRAND-SLAM software it came without any documentation – no licence, link to documentation or a 'getting started'/'readme'-type of document. This makes is a bit difficult to use. It's also provided as compiled java code, which is maybe not ideal for scientific software.

We apologize that the license text and readme (containing links to documentation) was indeed missing in the latest version. We now corrected this.

Specific points:

d. The dependencies need to be amended to include DESeq2 (which is used in some functions)

In R packages, dependencies can be defined in different ways. DESeq2 is defined as “Suggests” since it is only needed for specific tasks in grandR (differential gene expression testing), which is according to the best practices of R package development. However, a check for availability of DESeq2 before its usage was missing. We now carefully went through all function calls into “suggests” packages and added these checks, which now produce useful error messages asking to install the respective package.

e. SetParallel(cores = n) did not work for us (Xubuntu 20.04.3; R 4.1.1, 12 virtual cores). Could be an issue when working on a server.

We are not aware of any problems and use grandR frequently on a comparable server system. The only effect of using SetParallel is that afterwards, the mclapply function from the parallel package is used when appropriate. It is highly unfortunate, that this did not work for the reviewer, but we want to point out that for such kinds of problems of future users, we have an active eye on the bug-tracker on github.

f. It would be helpful if the gene names were included (e.g. as a title) in plots that showcase dynamics of individual genes, currently there does not seem to be an option to do this.

All plotting functions return ggplot objects, which has the advantage that titles and other elements can easily be added (e.g. PlotGeneOldVsNew(...)+ggtitle(“Title”)). We now made this more explicit in the plotting vignette.

The reason why we do not want to add this specific option is that there is a multitude of ways how users could want to adapt their plots (axis labels, where to put the legend, colors, other styling etc.), and we believe that using the mechanisms provided by the ggplot2 syntax provides the best flexibility while still being quite convenient.

g. Saying the workflow 'advocates' systematic sample names (first paragraph Results) is a bit of an understatement - the package does likely not work otherwise.

Here we respectfully disagree with the reviewer. Instead of defining the keywords to provide semantics for the parts of systematic sample names, users can also specify a “normal” metadata table. This is clearly mentioned in the help text of the respective functions in grandR and an example can be found in the corresponding vignette (which we believe is the right place for it). We made this now more explicit in the main text (“Our GRAND-SLAM – grandR workflow advocates (but does not enforce) using systematic sample names to encode all metadata”)

Minor comments

6. The synthesis and degradation rates are supposed constant, which is probably a good approximation. Can the model handle (slowly) varying rates?

This is indeed possible in a similar manner as for previous packages developed for metabolic labeling with affinity purification. With multiple snapshots over time, any change in kinetic rates would be approximated by a piecewise constant function in grandR. Figure 7 shows an example of varying rates. We now also added more discussion on this topic.

7. Some abbreviations (most notably NTR) are not formally introduced, decreasing readability

We thank the reviewer for this remark. Unfortunately, while revising our manuscript for the first submission, we inadvertently removed the definition of NTR. This is now corrected. Moreover, we carefully checked for other abbreviations that are not properly defined.

8. Results in Fig. 3a are described as 'significant' in the text, but I can't find p-values to back up the claim

Fig 3A shows empirical cumulative distribution of n=10,835 genes. P values (KS test) of all pairwise comparisons are $< 2.2 \times 10^{-16}$. This is now mentioned in text.

9. Ref. 14 is now published (Cao et al., Nature biotechnology, 2020)

Corrected.

10. Can the authors please add a reference about the active transport of 4sU across cell membranes?

We removed this misleading statement.

11. In the third paragraph of the Results, the authors states that 4sU treatment for > 4h is problematic, but the 4h threshold is only proven in the following abstract. This is a bit confusing to read, and the recommendation not to go beyond 4h should come after discussing Fig. 2c.

We removed our statement about a specific threshold (4h), since this depends on the cell type, 4sU concentrations and likely other conditions. We now phrased this conclusion more carefully: “Thus, our analyses here demonstrate the importance of assessing potential 4sU induced effects on transcription or RNA stability for pulse-chase designs and long-term labeling in general.”

Reviewer #3 (Remarks to the Author):

In the manuscript “GrandR: a comprehensive package for nucleotide conversions sequencing data analysis,” Rummel et al. address the experimental and computational challenges of studying the temporal dynamics of gene expression by metabolic RNA labeling and sequencing upon chemical

nucleotide conversion (i.e. SLAM-seq, Timelapse-seq or TUC-seq). To this end, the authors assembled a bioinformatic analysis workflow (**grandR**) that incorporates previously described computational methods (i.e., GRAND-SLAM) and well-established kinetic models to re-analyze previously published nucleotide conversion datasets.

While on the technical side, **grandR** may facilitate the analysis of nucleotide conversion sequencing data to measure cellular gene expression kinetics, the findings are limited in overall scope and do not provide substantial new technical or biological insights that would merit publication in *Nature Communications*. In fact, much of the parameters that are reported to impact the accuracy of mRNA half-live estimation strongly depend on the chosen experimental context (i.e., cell line, labeling conditions, library preparation protocol, global RNA metabolic state, etc.) and warrant careful experimental validation prior to drawing biological conclusions, an aspect that is rightly-so not addressed in this manuscript. The comments below should be considered by the authors to increase transparency and clarity of their workflow and benchmark the performance of **grandR** in light of substantially varying technical parameters using appropriately controlled experimental datasets.

We thank the reviewer for a careful consideration of our work. We would like to emphasize that the main purpose of our manuscript is to introduce a comprehensive software package for data analysis downstream of primary processing (as e.g. done by GRAND-SLAM). These downstream analyses represent a major hurdle in making use of nucleotide conversion sequencing approaches. We are therefore convinced that our software package will be of high value for many researchers.

Moreover, we disagree with the reviewer that our work does not provide substantial new insight. In addition to introducing a new software package and our simulation study comparing different models for estimating RNA half-lives, we would like to highlight:

- We show that it is insufficient to control toxicity effects by 4sU using cell viability assays, but we demonstrate that additional data quality control steps (compared to normal RNA-seq) are necessary to exclude effects of 4sU on expression estimates, which might otherwise lead to misinterpretations. Here, we propose a new method to perform quality control as part of **grandR**.
- We show that nominal labeling times can be quite different from effective labeling times, which impacts on half-live estimates, and propose and evaluate methods to recalibrate labeling times.
- We introduce a new method to estimate synthesis and degradation rates from snapshot experiments. We would like to emphasize that this is not limited to the mathematical transformation necessary to compute rate parameters from RNA abundances and the percentage of labeled RNA. Rather, we believe our main contribution here to be that our Bayesian method reports uncertainties in the kinetic parameters. Even if it is possible to estimate kinetic parameters from snapshots, the accuracy of those will be vastly different depending on how accurate the percentage of labeled RNA could be estimated, and depending on whether the labeling time matches the gene specific RNA half-life. Thus, gene-specific estimates of the measurement uncertainty in the rate parameters are of great importance.

Major comments:

For benchmarking of grandR, the authors build entirely on previously published datasets, that presumably vary in technical aspects, including library preparation protocol, cell type, and labeling

conditions (i.e. labeling efficiency and timepoints). But many of these technical parameters remain poorly addressed. How does grandR perform on datasets acquired in the same cell type but varying 4sU concentrations? In the absence of such a reference dataset, much of the presented analysis is rather circumstantial and does not allow to draw major conclusions. As a consequence, the statement that "these analyses advocate against long-term 4sU treatment beyond 4h even with low doses of 4sU and, therefore, argue against pulse-chase designs for estimating RNA half-lives in general" is misleading.

We would like to emphasize that the quoted statement is not the main conclusion of this part. Rather, we conclude the corresponding section saying that it is important to check for 4sU induced effects on transcription/RNA half-lives in such data as part of standard quality control, as e.g. implemented in our grandR package. Moreover, the statement quoted by the reviewer is not only based on our analyses of previously published data, but problems with long-term labeling with 4sU (e.g. rRNA processing defects) have been described earlier (Ref 20).

Nevertheless, the reviewer is right that we should not draw this conclusion in our manuscript based on the specific analysis done in this section. We therefore rephrased our statement as quoted by the reviewer accordingly: "Thus, our analyses here demonstrate the importance of assessing potential 4sU induced effects on transcription or RNA stability for pulse-chase designs and long-term labeling in general."

We agree that many technical factors such as 4sU concentration, cell type etc. impact on observed effects. However, the point of this small part of our manuscript is not to provide a comprehensive analyses of all possible factors and their combinations, but to (i) raise awareness of such issues, and (ii) provide a way to test for it.

Testing for a correlation of the RNA half-life (or the percentage of labeled RNA) vs. the log fold change between a 4sU labeled sample and a 4sU naïve sample has not been described before and we believe that this kind of quality control analysis, facilitated by our implementation in grandR, is an important contribution to the field.

The details on pre-processing steps, mapping parameters, chosen annotations etc. are largely missing in the methods section. In the absence of this information (and careful benchmarking) it is difficult to interpret the presented results.

We apologize that some details on preprocessing were missing in the manuscript. We now provide all parameters we used for each step of the preprocessing pipeline. The annotations we used were already clearly defined in the Methods section. Providing even more details on pre-processing steps would mean to detail the exact commands and scripts to perform the analysis. In fact, we do provide this on zenodo.

Furthermore, at least some of the analyzed data represent QUANTseq libraries (generated in at least two different cell types; mouse mESC, human MCF7) heavily rely on accurate annotation of mRNA 3' ends (which vary substantially between cell types), and preprocessing (i.e., trimming of random forward priming events, which may introduce errors in T>C conversion rate estimates). Do the presented data show per-gene or per-transcript 3' isoform analyses? If per-gene analyses, how was the signal averaged across different isoforms?

GRAND-SLAM generally provides gene-level analyses. The reported read count is the number of all reads that are compatible with at least one annotated isoform. The reported new-to-total RNA ratio is estimated by solving the binomial mixture model from all these reads. This is now clearly stated in

the Methods section. For all data analyzed, we trimmed the first 15bp from each read to remove artifactual mismatches induced by random priming as mentioned by the reviewer.

Using an annotation (Ensembl) that includes a comprehensive list of isoforms for all genes, and including each read mapping to any isoform mitigates the effects that cell type specific 3' ends might have on results. This is also the approach used by the de-facto standard for single cell data analysis, Cell Ranger, which generates (with regard to read locations) data that is very similar to Quant-seq.

However, GRAND-SLAM can also include custom annotations. In fact, the new analyses on the pulse-chase data (described below) were done on the 3' end clusters defined by Herzog et al. instead of the full Ensembl annotation (which was necessary in this case to make half-life estimates comparable to the ones published by Herzog et al.), and showed the same results (e.g., PCA) and let us draw the same conclusions.

While a long-term effect of 4sU labeling on gene expression (shown in Fig.2) is not entirely unexpected (given the reduction in cell viability to 80%), the effect on half-life calculation remains somewhat unclear. How do genes called differentially expressed at the 24h labeling timepoint (based on a very loose cutoff) vary in apparent expression along the entire timecourse? How do principal component analyses shown in Fig. 2A look like when including all labeling timepoints?

We now analyzed the full pulse-chase data set. We thank the reviewer for this suggestion to look at all chase time points. The PCA is now presented in Fig S1C. In short, the chase time points gradually move towards the no4sU samples. This suggests that in this experiment the 4sU induced effect on expression estimates is reversible.

On a single gene basis, these data are quite noisy:

These figures show volcano plots of each chase time point (including the 0h, i.e. 24h labeling) compared against the unlabeled control using total RNA. In each figure the same set of genes, namely the genes called differentially expressed in the 0h vs no4sU comparison (>2x up or down, multiple testing corrected P value <5%) is marked in red. Instead of analyzing individual genes, it is,

however, important to note, that the number of genes that are called differentially expressed (same criteria), gradually drops:

This is consistent with the PCA shown in the paper. We do not include these figures in the paper, since (i) they do not provide new insights and (ii) we feel that adding even more figures on the toxicity analysis would shift the emphasis too much to this part, which we consider important, but not the most important contribution of our manuscript.

Similarly, how does this analysis look like for all other published datasets? And how does half-life estimation change when excluding stepwise later labeling timepoints (particularly when investigating long-lived transcripts, for which accurate prediction relies on late measurement timepoints)?

For our toxicity analysis, we already consider two data sets (from Herzog et al. and Zuckerman et al.). For the Zuckerman et al. dataset we indeed did as the reviewer suggests here, exclude the last timepoint (Fig. S1D). However, as pointed out above, we do not want to add more emphasis on this part, and we also believe showing results from two independent experiments done with different designs (pulse-chase and progressive labeling) is sufficient for making our point, that such quality control tests are important.

While the analysis in Fig. S1C argues that short half-lived mRNAs (<5 h) are accurately modeled even in the absence of late measurement timepoints, this seems to not hold true for long-lived transcripts. Hence, the recommendations provided in the manuscript to measure (progressively) at early metabolic labeling timepoints would only hold for short-lived but not for long-lived transcripts. In fact, this is also observed in their simulated data (Fig. 3D) and hence, should be put in context.

We thank the reviewer for this very important remark. We now added this: “The variance in the differences between the estimated RNA half-lives with and without the 8h time point, however, were larger for long-lived RNAs. This is not surprising, since accurately estimating long RNA half-lives require long labeling^{16,23}. Thus, also for progressive labeling designs, labeling times in the order of 8h are beneficial, and 4sU concentrations should be chosen such that 4sU induced effects on RNA metabolism are minimized.”

The introduced analysis methods are tested/evaluated with in-silico datasets that were created with a simple simulation approach. It is, however, unclear how well these simulations reflect real data: how do simulated distributions of T-to-C conversions compare to real datasets? How does their method compare to other, established RNA-seq simulators (except for the missing T-to-C conversion feature)? Is their method producing realistic half-life distributions?

Reads were simulated according to an established RNA-seq simulator (Compcoder). The simulated RNA half-lives, read counts and overdispersions per gene as well as the global 4sU incorporation rate were matched to real data. As a consequence, the simulated distributions of T-to-C conversions also reflect this experiment. This is now made clear in the text.

The authors state in their introduction that “the percentage of such [i.e.4sU-labeled] reads is in the order of 20-80% and depends on the ratio of 4sU and normal uridine available for incorporation into nascent RNA, the read length, and the uridine content of RNA.” But in the presented re-analysis of published datasets and in their data simulations, the impact of these parameters on the performance of their analysis workflow is largely ignored. How does read-length (and U-content) influence robustness of their presented analysis workflow? What is the limit of 4sU-incorporation needed in order to draw meaningful biological conclusions? Such parameters seem as important for robust data acquisition as number of replicates or progressive timepoints.

The reviewer is correct that there are these other parameters that influence on the outcome of such experiments. However, these parameters only influence on how accurately the percentage of labeled (new) RNA (NTR) can be estimated by GRAND-SLAM. In our paper on GRAND-SLAM, this has been analyzed in detail. In short, e.g. low incorporation frequency leads to low accuracy, but this can be mitigated when there are more reads. Since these parameters belong to the realm of primary processing (i.e. GRAND-SLAM), and the number of possible and potentially interesting analyses and comparisons explodes due to the combinatorics, we here did not further expand on these parameters.

However, users can perform their own simulations using grandR, and it is indeed possible to define 4sU incorporation, read length and uridine to evaluate not only the effect of these parameters on the accuracy of the NTR estimation (as we did in the GRAND-SLAM paper), but also on downstream analyses such as half-lives estimates.

While the proposed temporal recalibration model obviously increases the model fit, it is less clear whether it results in more accurate half-life predictions. Did the authors compare corrected halflives to published/validated datasets or the mentioned ‘reference RNA half-lives’? The authors should also consider testing whether their proposed temporal recalibration model improves correlation of halflives between replicated data.

The “reference RNA half-lives” were mentioned in the text as follow: “For snapshot experiments, labeling times can be recalibrated based on additional assumptions, e.g. based on reference RNA half-lives”. We now made this clearer by adding: “that must be known a-priori. For the BANP data set, we made the assumption that globally, RNA half-lives are not affected by acute depletion of BANP, and therefore used the estimated half-lives of the untreated control sample as a reference.” Thus, we are not saying that there is a set of “reference RNA half-lives” that is available and that can be used as a gold standard to compare against. Comparing to a “published” data set also is prone to misleading conclusions, since all published RNA-half-lives have also been estimated based on data and models suffering from similar problems as mentioned in our manuscript.

Nevertheless, we the reviewer raised an important point. The only available data set that includes multiple experiments that can be used to estimate RNA half-lives for the same cells is the data from Herzog et al. This includes the pulse chase experiment, a Actinomycin D chase experiment, as well as several pulse time points that can be collected from separate experiments. We re-analyzed the pulse-chase as well as the pulse time points using GRAND-SLAM and estimated RNA half-lives for both using grandR with and without recalibration (for the pulse time points). Then we compared these to the half-lives estimated from either the pulse-chase or the Act-D data by Herzog et al. On

the bottom line, the correlations improved after recalibration, and also the absolute magnitudes of the half-lives were more similar after recalibration. These analyses are now shown in Figure 4C-D.

We furthermore somewhat disagree with the used formulation that for earlier timepoints the effective ‘labeling time’ is expected to be much shorter than the nominal time. Instead, we believe that the effective concentration of 4sU in the nucleus is increasing over time (until saturation) and thus the probability of a U being labeled increases over time. The authors should reformulate and test whether their assumption is true, e.g., by analyzing data from a 4sU chase labelling experiment in which cells are typically exposed to 4sU until full saturation. Or alternatively by providing an experimental estimate for labeling density. In this case they should not see the same effects on the residuals as for the pulse time course shown in Figure 4. They might also consider building such a gradual increase in effective 4sU concentration into their simulation approach to make it more realistic.

In our manuscript, we wrote: “Thus, the concentration of active 4sU increases until saturation, and RNA that was transcribed significantly before reaching saturation contains fewer 4sU than RNA transcribed later.” We therefore fully agree with the reviewer’s belief that “the effective concentration of 4sU in the nucleus is increasing over time (until saturation) and thus the probability of a U being labeled increases over time“. This means: Early during e.g. 2h of labeling, let’s say during the first hour, 4sU concentrations are low, and induce much fewer mismatches than later. Thus, in the first hour, there are so few mismatches that reads are not recognized as being labeled. In such a situation, the effective labeling time is 1h, which is shorter than the nominal labeling time of 2h. We explain this now better in the text by saying: “For example, the effective labeling time of a 1h sample might only be 40 minutes, since during the first 20 minutes, the concentration of activated 4sU might have been too low to induce many 4sU incorporation events.”

We analyzed the residuals of labeled RNA for the pulse-chase experiment of Herzog et al:

The later time points should be ignored, since the estimated labeled RNA for this is in most cases 0, whereas the model is only slightly above 0 (in which case the relative residual is -100%). Thus, as predicted by the reviewer, we do not see the same effects on the residuals. However, it is to be expected that also the 4sU washout is not instantaneous, i.e. that pulse-chase experiments suffer from similar problem (but apparently not that severely). Since this analysis does not provide new insights, we decided not to include it in the manuscript. Instead, we cite a recent preprint by the Churchman lab that provides convincing evidence for gradually increasing 4sU concentrations.

We indeed tested our recalibration approach using simulated data (Fig. S4).

Minor comments:

In the introduction the authors state that 4sU “is actively transported across cell membranes via nucleoside transporters.” Can the authors provide a reference for this statement? Prior work in yeast suggests that 4sU mainly enters cells via passive diffusion mediated by nucleoside equilibrate transporters of the Ent family.

We removed this misleading statement.

Introduction: “A limitation of 4sU conversion approaches is that concentrations of 4sU that are tolerated by cells commonly only replace 1 in 40 uridines by 4sU.” Where does this estimation come from?

In our GRAND-SLAM paper, we estimated the 4sU incorporation frequency to be between 2% and 2.5%. These estimates were corroborated by slightly lower (depending on the time of labeling) observed T-to-C mismatch frequencies within reads mapping to introns, most of which have an extremely short half-life such that virtually all observed intronic RNA is labeled (especially with long-term labeling). We added the reference.

NTR is being used inconsistently and is not well defined in the manuscript.

Unfortunately, while revising our manuscript for the initial submission, we inadvertently removed the definition of NTR. This is now corrected.

REVIEWER COMMENTS

Reviewer #1 (Remarks to the Author):

In this version of the manuscript, the authors add additional information on grandR, and provide more thorough discussions that make this manuscript clearer and more concise.

In particular, the authors expand the first part of the results section to provide a more comprehensive overview of grandR as a tool, and provide a cheatsheet summarizing the capabilities and use of grandR.

Although authors do not perform any direct comparison of grandR to other available tools, they do now discuss differences in capabilities, advantages and limitations of grandR in comparison to other available tools in the introduction and discussion.

The additional implementation of pulse-chase analysis in grandR is a relevant expansion, making the tool even more widely applicable.

However, some of the issues raised in the review have not been fully addressed.

The limitation of grandR in analysis of non-steady state situations is more explicitly discussed. However, analyzing the results of piecewise approximation by grandR, as the authors propose to use in such cases, in simulation studies of a temporally changing degradation and transcription would be a relevant addition to the manuscript.

“Choosing number of replicates, time points and sequencing depth”. Unfortunately, the analysis the authors perform does not answer the question raised in the review. Clearly, adding more samples would be beneficial. However, assuming we can only take 6 samples over an 8-hour time-course, which selection of samples would provide more accurate results? out of the two possible options:

(1) 2 x 0, 4, 8 hr (2 replicates at 3 time points as the authors already present)

- OR -

(2) 1 x 0, 1, 2, 4, 6, 8 hr (same number of samples, but no replicates, instead temporal resolution is higher).

Such an analysis would be very helpful to experimentalists trying to design sample collection within different systems.

We still find it challenging to follow all the different results presented in the manuscript, which are intermixed between validation of the grandR tool and general analysis related to experimental design of labeling etc.

Moreover, some figures and legends do not provide full details, making the interpretation of the results even more challenging. A few examples include:

(1) Figure 2B. x-axis has no title/description (not in figure nor in legend).

(2) Figure 2C. “NTR rank” (x-axis); should also mention the order of rank (lower to higher NTR).

(3) Figure 5F. Legend states both dotted and dashed lines. In figure, no dashed lines are plotted.

(4) Figure 6C. Y-axis titles should be located on left side (as conventionally done), and .

Therefore, most issues stated in our review questions have been addressed, and additional information has been included in the revised manuscript to make these points clearer. Some points have not been fully addressed.

Reviewer #2 (Remarks to the Author):

The authors have address all my comments in a fully satisfactory manner, and I don't have any further concerns. I support publication of the manuscript without further changes.

Reviewer #3 (Remarks to the Author):

In the revised version of the manuscript "grandR: a comprehensive package for nucleotide conversion sequencing data analysis", Rummel et al. have commented on major concerns raised in the original review. However, some aspects remain inadequately addressed and/or unclear:

The authors added further analysis on the long-term effect of 4sU labeling on gene expression by determining differentially expressed genes (analyses provided in a Fig not included in the manuscript but without reporting consistent changes of individual genes cross multiple timepoints). The authors come to the conclusion that "on a single gene basis, these data are quite noisy," which argues against a consistent effect of 4sU on gene expression. In case inconsistencies are observed, this should be reported. Merely reporting (inconsistent?) changes based on an arbitrary cutoff is not informative. The authors should at least transparently describe in the main section of the manuscript that the observed changes in gene expression are inconsistently observed across individual (and partially already early) timepoints and that technical reasons for such (minor) changes cannot be excluded. Again, the impact on half-life measurements (which is the main purpose of this technology) is not really addressed. The reader is still left with the impression that major changes in gene expression are induced even at low concentrations of 4sU (lowering this even further would jeopardize robust measurements), which is at odds with the later recommendation that "labeling times in the order of 8h are beneficial." Given the fact that the authors refrain from a systematic analysis that harmonizes possible (consistent) 4sU-dependent changes in gene expression with labeling efficiencies that enable robust detection of labeled transcripts, this aspect is important for the adequate usage of the method, even if the authors state that this is not the main purpose of their manuscript.

The reference given for the estimate of 2-2.5% labeling rates now refers to the GRAND-SLAM publication which describes a computational analysis pipeline. The authors should reference primary data that experimentally addressed incorporation rates (ideally with approaches other than RNAseq) and not merely their own publications.

REVIEWER COMMENTS

Reviewer #1 (Remarks to the Author):

In this version of the manuscript, the authors add additional information on grandR, and provide more thorough discussions that make this manuscript clearer and more concise. In particular, the authors expand the first part of the results section to provide a more comprehensive overview of grandR as a tool, and provide a cheatsheet summarizing the capabilities and use of grandR. Although authors do not perform any direct comparison of grandR to other available tools, they do now discuss differences in capabilities, advantages and limitations of grandR in comparison to other available tools in the introduction and discussion. The additional implementation of pulse-chase analysis in grandR is a relevant expansion, making the tool even more widely applicable.

We thank the reviewer for this very positive assessment.

However, some of the issues raised in the review have not been fully addressed.

The limitation of grandR in analysis of non-steady state situations is more explicitly discussed. However, analyzing the results of piecewise approximation by grandR, as the authors propose to use in such cases, in simulation studies of a temporally changing degradation and transcription would be a relevant addition to the manuscript.

We now implemented simulation of temporally changing degradation and transcription rates into grandR and used this to evaluate the constant rate approximation by grandR in a simulation study as suggested by the reviewer. Moreover, we believe it is important to theoretically investigate the nature of temporally changing rates and how the constant rate approximation behaves for different ways how rates might increase or decrease over time. In addition to our practical simulation study, we therefore now provide a Supplementary Note describing theoretical aspects of the constant rate approximation.

We added two new figures and the following text:

It is important to note that the kinetic model assumes constant rates of RNA synthesis and degradation during the time of labeling. If this assumption is not met, the estimated rates represent weighted averages of these varying rates within the labeling time (Supplementary Note 1). To test this, we again simulated data leaving one condition at steady state (and with constant rates) as control and let either the synthesis or degradation rate slowly approach a perturbed state over the time of labeling instead of setting them to a new value at the onset of labeling. The log fold changes vs control for the synthesis rates again reflected the true changes more accurately (RMSD=0.087, Fig 5G) than for the degradation rates (RMSD=1.609, Fig 5H). For both, the estimated log fold changes had a on average 80% lower magnitude than the true final rates after 2h of labeling corresponding to averaging over time for the estimated rates. We concluded that time-varying synthesis and degradation rates can be estimated using grandR, and that estimates correspond to weighted averages over the labeling time.

“Choosing number of replicates, time points and sequencing depth”. Unfortunately, the analysis the authors perform does not answer the question raised in the review. Clearly, adding more samples would be beneficial. However, assuming we can only take 6 samples over an 8-hour time-course, which selection of samples would provide more accurate results? out of the two possible options:

(1) 2 x 0, 4, 8 hr (2 replicates at 3 time points as the authors already present)

- OR -

(2) 1 x 0, 1, 2, 4, 6, 8 hr (same number of samples, but no replicates, instead temporal resolution is higher).

Such an analysis would be very helpful to experimentalists trying to design sample collection within different systems.

We would like to note that in addition to „2x 0,4,8hr“, as indicated by the reviewer also the option „1x 0,1,2,4,8hr“ had already been included (i.e. only lacking the 6hr time point compared to the reviewer’s recommendation). However, we appreciate that this particular comparison is interesting. We therefore use it as an introductory example of this part.

We still find it challenging to follow all the different results presented in the manuscript, which are intermixed between validation of the grandR tool and general analysis related to experimental design of labeling etc.

Moreover, some figures and legends do not provide full details, making the interpretation of the results even more challenging. A few examples include:

(1) Figure 2B. x-axis has no title/description (not in figure nor in legend).

(2) Figure 2C. “NTR rank” (x-axis); should also mention the order of rank (lower to higher NTR).

(3) Figure 5F. Legend states both dotted and dashed lines. In figure, no dashed lines are plotted.

(4) Figure 6C. Y-axis titles should be located on left side (as conventionally done), and .

(1) We added the x-axis label.

(2) We mention this now in the figure legend.

(3) Here we respectfully disagree. There are, and always were, dashed as well as dotted (and solid) lines:

(4) We respectfully disagree. „2h 4sU“ and „4h 4sU“ label the panels in the two respective rows (which is standard, and e.g. popularized by ggplot2 „facets“). The y axes are percentages („how many genes in a bin have changes in synthesis, half-life or both“), and are clearly described in the legend.

Therefore, most issues stated in our review questions have been addressed, and additional information has been included in the revised manuscript to make these points clearer. Some points have not been fully addressed.

We are confident that we now addressed all points raised by the reviewer with this revision and would like to thank the reviewer for his/her careful assessment and the highly useful recommendations.

Reviewer #2 (Remarks to the Author):

The authors have address all my comments in a fully satisfactory manner, and I don't have any further concerns. I support publication of the manuscript without further changes.

Reviewer #3 (Remarks to the Author):

In the revised version of the manuscript “grandR: a comprehensive package for nucleotide conversion sequencing data analysis”, Rummel et al. have commented on major concerns raised in the original review. However, some aspects remain inadequately addressed and/or unclear:

We apologize for aspects that we did not properly address in the previous revision and hope that the reviewer finds our responses to the remaining two concerns in this revision satisfactory.

The authors added further analysis on the long-term effect of 4sU labeling on gene expression by determining differentially expressed genes (analyses provided in a Fig not included in the manuscript but without reporting consistent changes of individual genes cross multiple timepoints). The authors come to the conclusion that “on a single gene basis, these data are quite noisy,” which argues against a consistent effect of 4sU on gene expression. In case inconsistencies are observed, this should be reported. Merely reporting (inconsistent?) changes based on an arbitrary cutoff is not informative. The authors should at least transparently describe in the main section of the manuscript that the observed changes in gene expression are inconsistently observed across individual (and partially already early) timepoints and that technical reasons for such (minor) changes cannot be excluded. Again, the impact on half-life measurements (which is the main purpose of this technology) is not really addressed. The reader is still left with the impression that major changes in gene expression are induced even at low concentrations of 4sU (lowering this even further would jeopardize robust measurements), which is at odds with the later recommendation that “labeling times in the order of 8h are beneficial.” Given the fact that the authors refrain from a systematic analysis that harmonizes possible (consistent) 4sU-dependent changes in gene expression with labeling efficiencies that enable robust detection of labeled transcripts, this aspect is important for the adequate usage of the method, even if the authors state that this is not the main purpose of their manuscript.

We apologize for not properly addressing this concern in the previous revision. Specifically, the volcano plots that we included as part of our reply are prone to a misunderstanding:

The genes called differentially expressed at 0h are shown as red dots in all vulcanos. This visualization gives the impression that they distribute randomly for all other time points. This is not the case, since genes are concentrated at the origin of the plot, and therefore a random distribution would shift the red dots predominantly also to the origin. A heatmap (now also part of the manuscript) shows that, overall, these changes are indeed quite consistent (colors are \log_2 fold changes vs the average of the no4sU replicates):

Importantly, this is the same data as shown in the vulcanos, they are only visualized differently.

Moreover, our new analyses now also assess the impact on half-life estimates by comparing the 4sU-pulse-chase estimated half-lives to half-lives estimated from an experiment not involving 4sU (Act-D chase). This analysis indicated that gene expression changes are due to transcriptional regulation, i.e. the half-lives estimated from 4sU-pulse-chase are unbiased even for genes with changed expression.

We rewrote this part in our revised manuscript. Specifically:

- We changed the \log_2 FC cutoff of the volcano plot from 0.5 to 1 (i.e. highlighting genes that are >2x regulated). The intention of the 0.5 cutoff was to provide more evidence that there is a global, but mild effect on gene expression. The PCA plot also shows this global effect but does not provide insight into the extent of the changes in gene expression. With changing the cutoff to 1 for the volcano plot, we believe to prevent the potential misinterpretation that a lot of strong changes occurred.
- We removed the analysis of the correlation with the RNA half-life since this was hard to interpret and did not provide new insights.
- We added heatmaps to provide a global overview on expression of (i) regulated genes, (ii) of all other genes and (iii) of data from shorter labeling pulses.
- We compare half-lives from 4sU-pulse-chase experiments against Actinomycin-D experiments to show that up- and down-regulation is not due to a change in RNA stabilities

In the following we will respond to key sentences written by the reviewer:

but without reporting consistent changes of individual genes cross multiple timepoints

We now report consistent changes in Fig 2B.

that “on a single gene basis, these data are quite noisy,” which argues against a consistent effect of 4sU on gene expression

We apologize that our phrasing “quite noisy” in our previous response was unfortunate and prone to misunderstanding. What we meant was that a gene that reached statistical significance for one time

point does not necessarily reach significance in other time points. We would like to emphasize that not reaching statistical significance, especially with low statistical power due to only three replicates, does not allow to draw any conclusions, and should not be called “inconsistent”.

We agree with the reviewer that the data as presented in our previous response (vulcanos) did not argue for a consistent effect. We are confident that the heatmap visualization provides a compelling argument that changes are indeed consistent.

Again, the impact on half-life measurements (which is the main purpose of this technology) is not really addressed. The reader is still left with the impression that major changes in gene expression are induced even at low concentrations of 4sU

With our new analyses we now address this. The presented data clearly show that there are changes in gene expression that are induced at low 4sU concentrations, and they are not just technical artifacts that are inconsistently observed. Importantly, this is not the first time that effects of low dose 4sU concentration are described (e.g. Burger et al., RNA Biol 2013). However, we believe that we make a strong argument that despite that RNA half-lives from the pulse-chase experiment are accurate.

The reference given for the estimate of 2-2.5% labeling rates now refers to the GRAND-SLAM publication which describes a computational analysis pipeline. The authors should reference primary data that experimentally addressed incorporation rates (ideally with approaches other than RNAseq) and not merely their own publications.

We changed this to the SLAM-seq publication of Herzog et al., where labeling rates were measured using both sequencing and mass spectrometry.

REVIEWERS' COMMENTS

Reviewer #1 (Remarks to the Author):

In this revision the authors have addressed all the remaining points, and I do not have any additional concerns.

I would still like to note that the problem regarding the graphics of figure 5F was not resolved.

In the PDF version provided to the reviewers both in the previous and also in the current revision, there are no dashed lines evident in the figure.

However, I now understand that this is likely a result of conversion issues during PDF generation, and therefore urge the authors to verify this point prior to publication.

Reviewer #3 (Remarks to the Author):

The authors have now provided a transparent assessment of the data, as inquired. Minor comment: There are display problems in the revised versions of Figs 2D, 5H,I and S1B.

REVIEWERS' COMMENTS

Reviewer #1 (Remarks to the Author):

In this revision the authors have addressed all the remaining points, and I do not have any additional concerns.

I would still like to note that the problem regarding the graphics of figure 5F was not resolved.

In the PDF version provided to the reviewers both in the previous and also in the current revision, there are no dashed lines evident in the figure.

However, I now understand that this is likely a result of conversion issues during PDF generation, and therefore urge the authors to verify this point prior to publication.

We noticed two issues that occurred with different versions of the default pdf viewer ("preview") of MacOS only. Since we only checked under linux using evince as well as under MacOS using Adobe Acrobat Reader, we did not spot these issues ourselves in the first place. We solved both issues and checked all Figures under Windows, Linux, MacOS with different programs (including the MacOS preview tool).

One issue occurs when using dashed or dotted with the `stat_ecdf` function from the `ggplot2` package (this can be circumvented by precomputing the ecdf and then using `geom_line`), the other is related to rasterization of point layers using the `ggrastr` package and then using the program "inkscape" to convert the layouted figure into a pdf (this can be circumvented by using `rsvg-convert` for the conversion).

Reviewer #3 (Remarks to the Author):

The authors have now provided a transparent assessment of the data, as inquired. Minor comment: here are display problems in the revised versions of Figs 2D, 5H,I and S1B.

See above!